# Functionalization of Octacalcium Phosphate Bone Graft with Cisplatin and Zoledronic Acid: Physicochemical and Bioactive Properties

**DOI:** 10.3390/ijms241411633

**Published:** 2023-07-19

**Authors:** Ekaterina A. Kuvshinova, Nataliya V. Petrakova, Yulia O. Nikitina, Irina K. Sviridova, Suraja A. Akhmedova, Valentina A. Kirsanova, Pavel A. Karalkin, Vladimir S. Komlev, Natalia S. Sergeeva, Andrey D. Kaprin

**Affiliations:** 1P.A. Herzen Moscow Research Oncology Institute, Branch of FSBI National Medical Research Radiological Centre, Ministry of Health of the Russian Federation, 2nd Botkinsky Pass. 3, 125284 Moscow, Russia; i.k.sviridova@yandex.ru (I.K.S.); prognoz.06@mail.ru (S.A.A.); kirik-57@mail.ru (V.A.K.); pkaralkin@gmail.com (P.A.K.); prognoz.01@mail.ru (N.S.S.); 2A.A. Baikov Institute of Metallurgy and Materials Science RAS, Leninsky Avenue 49, 119334 Moscow, Russia; nyo.94@yandex.ru (Y.O.N.); komlev@mail.ru (V.S.K.); 3L.L. Levshin Institute of Cluster Oncology, I.M. Sechenov First Moscow State Medical University, Trubetskaya 8-2, 119991 Moscow, Russia; 4FSBI National Medical Research Radiological Centre, Ministry of Health of the Russian Federation, 2nd Botkinsky Pass. 3, 125284 Moscow, Russia; 5Department of Urology and Operative Nephrology, Peoples’ Friendship University of Russia, Miklukho-Maklay Str., 6, 117198 Moscow, Russia

**Keywords:** octacalcium phosphate, cisplatin, zoledronic acid, functionalization, kinetics of drug release, osteoinductivity, cytostatic and antitumor properties

## Abstract

Bones are the fourth most frequent site of metastasis from malignant tumors, including breast cancer, prostate cancer, melanoma, etc. The bioavailability of bone tissue for chemotherapy drugs is extremely low. This requires a search for new approaches of targeted drug delivery to the tumor growth zone after surgery treatment. The aim of this work was to develop a method for octacalcium phosphate (OCP) bone graft functionalization with the cytostatic drug cisplatin to provide the local release of its therapeutic concentrations into the bone defect. OCP porous ceramic granules (OCP ceramics) were used as a platform for functionalization, and bisphosphonate zoledronic acid was used to mediate the interaction between cisplatin and OCP and enhance their binding strength. The obtained OCP materials were studied using scanning electron and light microscopy, high-performance liquid chromatography, atomic emission spectroscopy, and real-time PCR. In vitro and in vivo studies were performed on normal and tumor cell lines and small laboratory animals. The bioactivity of initial OCP ceramics was explored and the efficiency of OCP functionalization with cisplatin, zoledronic acid, and their combination was evaluated. The kinetics of drug release and changes in ceramics properties after functionalization were studied. It was established that zoledronic acid changed the physicochemical and bioactive properties of OCP ceramics and prolonged cisplatin release from the ceramics. In vitro and in vivo experiments confirmed the biocompatibility, osteoconductivity, and osteoinductivity, as well as cytostatic and antitumor properties of the obtained materials. The use of OCP ceramics functionalized with a cytostatic via the described method seems to be promising in clinics when primary or metastatic tumors of the bone tissue are removed.

## 1. Introduction

Bones are the fourth most frequent site of metastasis from malignant tumors, including breast cancer, prostate cancer, melanoma, etc. [1]. The treatment of patients with bone metastases is usually a complex process that involves chemotherapy, radiation therapy, and surgery.

In cancer patients, the insufficient regenerative capacity of bone tissue limits its rehabilitation after surgical intervention [2,3,4]. In these cases, the implantation of synthetic calcium phosphate (CaP) materials is aimed at stimulating bone regeneration due to their osteoconductive and osteoinductive potential. The chemical composition, porosity, specific surface area (SSA), and bioresorbability are the key factors for effective regenerative processes [5,6,7].

On the other side, the low bioavailability of bone tissue for drugs makes chemotherapy ineffective in patients with bone tumors. With this, the escalation of therapeutic doses of drugs leads to pronounced toxic side effects [8,9,10]. In this regard, the development of methods for the functionalization of CaP materials with anticancer drugs is aimed at their target delivery to the pathology site for the local release of therapeutic drug concentrations [11,12,13,14,15,16,17,18,19,20,21].

The effectiveness of drug incorporation into the CaP carrier, the rate, and duration of its release depend on the type of interaction with the material. With chemical adsorption via covalent or ionic binding to material surface, a given amount of the drug can be incorporated into the CaP carrier [9,22,23]. However, the rate of its release may be insufficient to achieve an effective concentration in the defect zone. With physical adsorption, binding occurs due to electrostatic interactions, van der Waals forces, and hydrogen bonds [24,25,26]. These bonds are reversible and do not lead to changes in drug activity. However, bindings are characterized by low strength, and the drug release kinetics often have an initial burst release phase [27,28]. Hence, when developing an effective method for CaP material functionalization with anticancer drugs, the structure and composition of the carrier and the drug, as well as the nature and strength of their binding, must be taken into consideration.

In this work, octacalcium phosphate (OCP) bone graft in the form of ceramic porous granules (OCP ceramics) was used as a platform for delivering cisplatin to a bone defect. Among synthetic materials developed for bone reconstruction, OCP ceramics are considered to be the most preferable bone graft substitute due to their outstanding physicochemical and bioactive properties. During bone mineralization, the formation of bone apatite occurs through nucleation and the formation of hydroxyapatite (HA) precursors within the organic matrix of bone [29,30]. OCP is regarded as one of the forms of bone apatite precursor [31]. In an aqueous environment, OCP undergoes chemical conversion via hydrolysis to HA, accompanying calcium ion absorption and phosphate ion release [32]. The ionic exchange processes, which are attendant to OCP hydrolysis, increase the resorption rate of the material and, furthermore, indirectly stimulate osteogenic differentiation [33,34,35,36]. The chemical feature of OCP implied by the two-layer structure of apatitic and hydrated layers determines the inherent OCP crystal morphology and high specific surface area, which ensures its pronounced adsorption properties. These key features provide OCP with pronounced osteoinductive properties [34,37,38,39,40,41].

Cisplatin (Cis), the most frequently used cytostatic in the chemotherapy of primary and metastatic bone tumors, was applied for OCP ceramic functionalization. The cytostatic effect of this drug is based on contraventions of the structure, synthesis, and repair of DNA, leading to a halt in the proliferation of malignant cells [42].

A review of the relevant literature showed that there are very few studies on the incorporation of cisplatin into CaP materials. Some works show the methods to slow down the release of Cis from the CaP material: the use of bulk (rather than granular) implants [43] or composite materials [44], variations in the phase composition of the CaP materials [19], and the use of vacuuming during the functionalization procedure [19]. Other studies revealed the cytostatic properties of CaP materials functionalized with Cis and its derivatives. For instance, nanocrystalline HA containing phosphoplatin inhibited the in vitro proliferation of tumor cells Hela, MCF-7 [45], and A549 without loss of osteoinductive properties [46].

In this work, a new technique for OCP functionalization with cisplatin was developed through the combined incorporation of the cytostatic with a bisphosphonate (BP)—zoledronic acid (Zol). It is well known that BPs are inhibitors of osteoclast activity and have a direct apoptotic effect on tumor cells. Therefore, BPs are frequently included in the complex therapy of osteolytic tumor metastases in bone tissue [11,47,48,49,50].

Some approaches to functionalization of osteoplastic materials with BPs were described mainly through saturation from solutions or coprecipitation with CaP [49,51,52,53,54]. In addition, the interaction of BPs with CaP occurs due to chemisorption onto its surface. During interaction, BPs form coordination complexes with calcium ions of CaP when negatively charged phosphonic groups covalently bind with metal ions via oxygen atoms. Hydroxyl groups and nitrogen-containing groups of BP enhance its binding to the carrier by forming electrostatic bonds with OH groups of CaP [11,23].

Considering the BPs’ binding ability to CaPs, we adopted the approach of employing BP Zol as a mediator of interaction between Cis and OCP. Significant slowing down in Cis release from OCP ceramics and gradual release profile confirmed the enhancement of their binding strength via Zol intermediation.

Thus, the purposes of this study were to investigate the functionalization of OCP ceramics with cytostatic Cis, BP Zol, and their combination (Cis/Zol), to study the kinetics of their release from the material and to assess the physicochemical and bioactive properties of the obtained materials. The presented results show that the developed method of OCP ceramic functionalization with cisplatin in combination with zoledronic acid makes it possible to obtain an osteoplastic material with osteoinductive and antitumor properties, which are preserved for a long time. The study demonstrates the unique results on Cis release from functionalized ceramics: only half of the Cis-incorporated amount was released within 5 weeks. It has also been shown that the functionalization with a combination of Cis and Zol provides a long-term cytostatic effect against the breast cancer cell line. In vivo experiments showed the advantages of targeted delivery of cisplatin to the tumor growth zone using the developed material in comparison with an intravenous injection of this cytostatic agent. Thus, the developed method of OCP ceramic functionalization will ensure the prevention of tumor recurrence at the site of bone tissue tumor removal while stimulating its regeneration.

## 2. Results

### 2.1. OCP Ceramic Characterization

The results of the structure study using scanning electron microscopy (SEM) showed that the initial (drug-free) ceramic granules were porous and characterized by an average size of 500–1000 μm and irregular shape (Figure 1A–C). The macropores occurred from the replica fabrication method, burning out the organic foam, at about 50–100 μm. Micropores of 10–20 μm were formed due to contact of the OCP crystals composing the material. Elongated plate-like OCP crystals were 5–10 μm in width and about 30 μm in length, assembled in flower-like aggregates.

The Fourier-transform infrared (FTIR) absorption spectrum of the material confirmed the formation of well-crystallized OCP (Figure 1D). The characteristic absorption bands of phosphate groups (PO_4_) were represented by a set of bands corresponding to P-O stretching vibrations at 962 and 1020 cm^−1^ (*v*_1_, *v*_3_) and to P-O bending vibrations at 467, 559, and 600 cm^−1^ (*v*_2_, *v*_4_) [55,56]. The presence of HPO_4_ (HOPO_3_) groups is an inherent structural feature of OCP. These are mainly represented by two structurally different groups, each having one OH stretching and OH bending modes [57,58]. The FTIR spectrum of the material contained a set of characteristic modes of HPO_4_ groups: weak HPO_4_ bend (*ν*_2_), two in-plane OH bends at 1182 and 1295 cm^−1^, and two bands at 917 and 860 cm^−1^ assigned to the P-(OH) stretching vibrations of the HPO_4_ group.

The X-ray diffraction (XRD) data indicated that the composition of the granules corresponded to the OCP phase (Figure 1E). The SSA of OCP ceramics was 29.9 ± 0.9 m^2^/g.

### 2.2. Bioactivity Study of OCP Ceramics

The bioactivity study included a study of cytotoxicity and cytocompatibility of OCP ceramics on MG-63 osteoblast-like cells and a study of osteoinductive properties of OCP-ceramics on human bone marrow mesenchymal stem cells (BM MSCs).

The MTT assay revealed the absence of cytotoxicity in OCP ceramics in indirect contact with MG-63 cells. The population of viable cells (PVCs) was 71.0 ± 0.3% in comparison with untreated culture (control) (data shown in Appendix A). However, in the direct contact, the dynamics of the MG-63 cells increase lagged behind the control during observation periods of up to 10 days, and by day 14, they equaled (Figure 2A). Culturing MG-63 cells with OCP in a complete growth medium (CGM) caused a decrease in Ca^2+^ content and an increase in phosphorus (Pi) content in the medium after 5 days of observation. These indirectly indicated OCP partial hydrolysis to HA (Figure 2B) [59]. Probably, it caused a delay in MG-63 cell proliferation. Further, after 10 days of culturing, the ion exchange processes stabilized; along the surface, OCP hydrolyzed to HA, which made it more attractive to cells (Figure 2A).

OCP ceramics induced the differentiation of macrophages into osteoclasts, increasing the expression of the *TRAP* marker gene by 1.7 ± 0.1 times compared with the *TRAP* expression in untreated macrophages (control) (Figure 3). The expression of this gene was increased by 5.8 ± 0.2 times with addition of the inducing factor RANKL. At the same time, *TRAP* expression in RAW 264.7 macrophages cultured in direct contact with OCP ceramics and with the addition of RANKL was 15.8 ± 0.6 times more intense than in the control (Figure 3).

The culturing of human bone marrow mesenchymal stem cells (BM MSCs) in direct contact with OCP ceramics for 14 days caused an increase in the expression of the *RUNX2* and *SP7* genes in these cells by 6.3 ± 1.1 and 29.2 ± 0.7 times compared with the expression in untreated BM MSCs (control) (Figure 3). In fact, these changes were even more pronounced than in cells subjected to directed differentiation using the osteogenic medium (OM) StemPro Osteogenesis Differentiation Kit, in the presence of which the expression of these genes was 3.7- and 2.1-times higher than in the control, respectively (Figure 3). Thus, OCP ceramics induced the osteoblastic differentiation of BM MSCs.

### 2.3. Functionalization of OCP Ceramics with Cisplatin

During the functionalization procedure of OCP granules with Cis water solution, the drug adsorption onto the OCP surface took place. While the Cis concentration was 1 mg/mL, the amount of Cis contained in OCP granules increased in the first two days of the material incubation in an aqueous Cis solution. Further material incubation caused a decrease in the Cis content in the functionalized granules (Figure 4A). The Cis water solution of pH = 5.5–5.9 provoked a slight degradation in the ceramic surface with a crystal reduction (Figure 5). An increase in the incubation time (up to 7 days) led to an intensification of degradation processes, accompanied by the release of the drugs.

The amount of Cis adsorbed on OCP granules increased as a linear function of Cis concentration in an incubation solution from 0.5 to 2.0 mg/mL (Figure 4B). The sharper slope angle of concentration function indicated a gradual saturation of the OCP surface with the drug.

The kinetics of drug release from OCP ceramics, functionalized with Cis (OCP-Cis), were non-linear (Figure 4C). In the first hour and in 1 day, 34.7% and 72.5% of the incorporated Cis was released from the material, respectively (initial Cis concentration—1 mg/mL). The curve reached a plateau after 7 days and, during this time, almost all of the incorporated Cis was released (98.6%).

### 2.4. Functionalization of OCP Ceramics with Zoledronic Acid

Figure 6 represents the effectiveness of Zol incorporation into OCP granules using Zol-incorporative solutions of various concentrations and an incubation time of 2 days (OCP ceramics functionalized with Zol were designated as OCP-Zol). The dependence of the amount of incorporated Zol on its concentration in the incorporative solution corresponded to a linear function, which indicates the absence of the saturation of ceramics with this drug (Figure 6A).

At the same time, SEM study of the material surface revealed evidence of crystal degradation as a result of functionalization with solutions of higher Zol concentration (more than 1.0 mg/mL). Zol water solutions were characterized by acidic properties (pH values = 2.0–4.0), which led to dissolving and reducing the surface crystals of OCP (Figure 7). In addition, SEM study revealed the presence of a new phase of rod shape, with a cross-section up to 0.5 μm (Figure 8). The content of the rods increased with Zol concentration in the incorporative solution. Apparently, the derivative of zoledronic acid, calcium zoledronate, was formed due to chemical interaction between bisphosphonate and calcium ions released from OCP during degradation while incubated in the Zol solution. A similar formation of the rod-shaped product of Zol and OCP interaction was demonstrated by Forte L. et al. and Boanini E. et al. [49,60]. Figure 8D presents an XRD pattern of the Zol rods pulled up from the vessel with OCP and incorporative Zol solution of 4 mg/mL after 2 days of incubation. The determination of the powder XRD patterns for the Zol rods did not show the exact indexation of the product. Meanwhile, the diffraction peaks at 2Theta = 6.5 and 9 deg., like the major XRD lines for the Zol rods in Figure 8D, were revealed for the product of incubation of β-tricalcium phosphate (β-TCP) with increasing quantities of Zol (0.01–0.08 mol/L) in [61]. The authors found that it was the mixed sodium and calcium Zol complex, resulting from a partial dissolution of β-TCP.

Zol release from OCP-Zol ceramics occurred in insignificant quantities (Figure 6B). During 35 days of the observation, 6.3 and 9.3% of Zol were released from the material functionalized with solutions of 0.5 and 1.0 mg/mL of Zol, respectively. In the kinetics of Zol release, a short phase (1 h) of burst release was also observed (Figure 6C). During this time, only 3.8% and 4.6% of the incorporated Zol were released (concentrations in the incorporative solution of 0.5 and 1.0 mg/mL, respectively).

### 2.5. Functionalization of OCP with Cisplatin and Zoledronic Acid

The functionalization of OCP ceramics with cisplatin in combination with zoledronic acid was carried out using incorporative solutions with drug ratios of 1:0, 1:0.2, 1:0.5, and 1:1, respectively (the obtained ceramics were designated as OCP-Cis-1, OCP-Cis/Zol-1:0.2, OCP-Cis/Zol-1:0.5, OCP-Cis/Zol-1:1, respectively). In the result, the proportion of Cis released from OCP over 5 weeks in relation to incorporated Cis decreased from 96% (pure Cis) to 35.1% (Cis: Zol 1:1) (Figure 9A). When recalculated in absolute values, after 5 weeks, in the absence of Zol, 0.43 μg/mg of Cis remained bound to OCP, and in Zol presence—from 2.19 to 5.04 μg/mg (for OCP-Cis/Zol-1:0.2, OCP-Cis/Zol-1:0.5, and OCP-Cis/Zol-1:1). In addition, when Cis was incorporated with Zol, it was released more smoothly, even in the first few hours after incorporation (Figure 9B).

In general, the kinetics of the release of Zol and Cis from OCP were fundamentally different (Figure 4C and Figure 6B,C). The amount of released Zol was an order of magnitude less than that of Cis, even at its maximum initial concentration. In addition, after 7 days of observation, the release of Zol practically ended, while the release of Cis continued throughout the observation period (5 weeks). Thus, the specific therapeutic properties of OCP ceramics functionalized with both drugs will most likely be determined by the effect of the released cytostatic, since its concentration in the environment surrounding the material will be an order of magnitude higher than the BP concentration.

### 2.6. In Vitro Study of Degradation of OCP and OCP-Zol Ceramics

It was found that soaking drug-free ceramics to SBF was accompanied by the adsorption of calcium ions and the release of phosphate ions, especially intensively on the seventh day of the study (Figure 10A,B). Further soaking resulted in a decreasing ion exchange between ceramics and model solution up to the 56th day. The weight loss of the samples lasted up to 14 days of soaking (up to 3%); then, a slight weight rise occurred (Figure 10C). It is obvious that the hydrolysis of OCP to HA began from the first periods of exposure; later (days 3–7), this process was accompanied by the formation of a new CaP phase precipitated from a saturated calcium phosphate buffer solution, as we showed earlier [59].

The incorporation of Zol into OCP ceramics reduced the intensity of all indicated processes (Figure 10). In materials previously saturated with Zol solution (concentration of 0.5 and 1 mg/mL), the calcium adsorption and phosphorus release were less intensive. On day 14 of exposure, materials saturated with 0.5 mg/mL Zol solution adsorbed 15% less of [Ca^2+^] ions from SBF than drug-free OCP; materials saturated with 1 mg/mL Zol solution—32.5% less. Phosphate ions were not released into the model solution. When exposed to SBF, OCP-Zol ceramics increased their mass only in the first 3 days (by about 1–2%), followed by a mass loss.

Figure 10D shows FTIR spectra of the materials at the most indicative terms of exposure to SBF (7 and 28 days). The intensity of bands specific to HPO_4_ groups of OCP decreased with an increase in the time of exposure to SBF, indicating a gradual hydrolysis of OCP to HA. However, no bands of OH groups specific to the HA structure were detected (at 560 cm^−1^), indicating only partial OCP hydrolysis during exposure to the model solution.

According to the SEM study, the formation of a new CP phase on the surface of drug-free OCP ceramics took place during exposure to SBF. The quantity and distribution of the latter phase increased with a prolonged exposure time (Figure 11A). The presence of the newly formed CP phase as needles of low crystallinity on the primary OCP plate-like crystals was confirmed by many studies [59,62,63,64,65] and was mainly attributed to calcium-deficient HA. Its formation was caused by OCP hydrolysis and subsequent CaP precipitation from the saturated SBF solution. The surface of Zol-OCP also contained the newly formed CP coating as a result of precipitation, with slightly different morphology on the early exposure terms. Increasing the time of exposure to SBF led to the appearance of rounded particles aggregated in thin films and located between clusters of large OCP crystals (Figure 11B,C). The content of the rounded particles increased with Zol concentration in OCP ceramics. Their distribution on the surface of the OCP ceramics was irregular and counted as an insignificant part of the total material volume; therefore, performing XRD indexation of this phase was irrelevant (Appendix A). We attribute the appearance of these structures to the effect of BP on the morphology of the CP coating during precipitation in SBF. Probably, the rods of calcium zoledronate were precipitated as a result of the interaction of Zol with cations, releasing from OCP during its partial dissolution in incorporative Zol solution. Thereafter, with exposure to SBF, calcium zoledronate was soluble in solution and bound with accompanying ions of SBF. The different morphology of crystalline Zol particles depending on cation nature was shown in [66]. In this way, the formation of the particles in the form of thin films is associated with the interaction of zoledronate with metal ions in the SBF solution.

Thus, it can be concluded that during saturation with Zol, it chemically binds to OCP crystals, affecting the kinetics of its degradation in SBF. The adsorption of calcium ions from the solution and the release of phosphate ions were less intensive compared to drug-free OCP. This effect is enhanced with an increase in the concentration of bisphosphonate, indicating the inhibition of the processes of OCP dissolution and hydrolysis.

### 2.7. Effect of OCP-Zol on the Differentiation of Osteogenic Cells

It is known that BPs affect osteoclasts, both by activating their apoptotic death and by inhibiting the processes of differentiation of their precursors (i.e. macrophages) [47,48]. Therefore, in the first stage, we assessed the sensitivity of RAW 264.7 macrophages (intact and differentiated into osteoclasts by the RANKL factor) to Zol. In accordance with the obtained dependence, the dose–effect IC_50_ for undifferentiated cells (macrophages) was 9.74 µg/mL, and for differentiated cells (osteoclasts), it was 14.77 µg/mL (Figure 12A). Therefore, to assess the effect of Zol on the differentiation of RAW 264.7 cells, non-toxic Zol concentrations of 0.025 and 0.25 µg/mL were chosen.

At the indicated concentrations, Zol significantly inhibited RANKL-induced differentiation of macrophages into osteoclasts: fold difference (FD) in *TRAP* expression in the experimental culture compared to the *TRAP* expression in the control culture (untreated RAW 264.7 cells) decreased from 10.2 ± 0.9 (for RANKL) to 2.6 ± 0.4 and 4.8 ± 0.8 (for the combination of RANKL with Zol at concentrations of 0.025 and 0.25 µg/mL, respectively) (Figure 12B). These data confirm the adequacy of the model used.

As shown above, OCP induces the differentiation of macrophages into osteoclasts, as evidenced by the expression of *TRAP* in their precursor cells RAW 264.7. At the same time, OCP functionalized with Zol at its concentrations of 0.2, 0.5, and 1.0 mg/mL of the incorporative solution (OCP-Zol-0.2, OCP-Zol-0.5 and OCP-Zol-1.0, respectively) did not induce RAW 264.7 differentiation; when RAW 264.7 cells were cultured in the presence of OCP-Zol, *TRAP* expression levels were lower compared to the control value (FD 0.61 ± 0.1, 0.42 ± 0.02, and 0.99 ± 0.1 for OCP-Zol-0.2, OCP-Zol-0.5, and OCP-Zol-1.0, respectively, control—untreated RAW 264.7 cells) (Figure 13A).

Moreover, this effect was most pronounced in the material with a lower content of Zol (OCP-Zol-0.2 and OCP-Zol-0.5). Thus, Zol blocked the osteoclast differentiation-inducing effect of OCP.

Culturing human BM MSCs in the presence of OCP ceramics did not cause an increase in *ALPL* expression (FD < 1) but significantly increased *RUNX2* expression (Figure 13B). OCP ceramics functionalized with Zol at different concentrations (0.2–1.0 mg/mL in incorporative solution) increased the expression of both *ALPL* and *RUNX2* in BM MSCs. The highest level of gene expression was observed at a Zol concentration of 0.5 mg/mL (Figure 13B). Therefore, the introduction of Zol in OCP enhances its osteoinductive properties.

### 2.8. Assessment of Cytostatic Properties of Functionalized OCP Ceramics

An in vitro study of cytostatic properties of OCP-Cis, OCP-Zol, and OCP-Cis/Zol was performed on MCF-7 breast cancer cells. First, the correctness of the experimental model by studying MCF-7 cell sensitivity to the selected drugs in a concentration range from 0.01 to 100.0 µg/mL was confirmed.

The cytostatic effect of Cis on MCF-7 tumor cells was demonstrated. The IC_50_ value for Cis was 7.45 µg/mL. However, Zol did not exert a pronounced inhibitory effect on tumor cells in the studied concentration range (Figure 14A).

Figure 14B demonstrates the cytostatic effect of functionalized OCP on MCF-7 cells. When studying the cytostatic effect of functionalized OCP on MCF-7, the culture medium was not changed for 7 days, which ensured the accumulation of drugs released from OCP in the medium. In the group with drug-free OCP in the culture medium, the population of viable cells (PVCs) decreased to 67.0 ± 14.4% by day 7 of cell growth, which was probably due to the depletion of the nutrient medium (Figure 14B).

The presence of OCP-Cis in the culture medium led to an increase in the cytostatic effect: PVC decreased from 68.6 ± 5.5% on the first day of culturing to 3.2 ± 1.0% on the seventh day (Figure 14B). OCP-Cis/Zol had a less pronounced cytostatic effect. Moreover, the greater the functionalizing concentration of Zol with an equal concentration of Cis in the incorporative solution, the lower the cytostatic effect (Figure 14B). It was likely conditioned by the decrease in the Cis quantity bound to OCP and by slowing down its release with an increase in Zol concentration in the initial solution.

In the next step, the viability of MCF-7 tumor cells cultured for 7 days in the presence of functionalized ceramics pre-soaked in CGM (5, 12, and 19 days) (Figure 14C) was evaluated. Hence, the incorporated drugs were partially released from the samples, and only the drugs remaining in the ceramics affected the cells.

This experiment simulated the conditions in the bone defect arising from the implantation of functionalized materials. Indeed, under in vivo conditions, the drugs released from the material are washed out of the defect area by the body’s interior. Consequently, the long-term antitumor effect of the material can be ensured only by new portions of continuously released drugs.

It was found that after 5 days of pre-soaking in CGM, OCP-Cis-1, OCP-Cis/Zol-1:0.2, OCP-Cis/Zol-1:0.5, OCP-Cis/Zol-1:1, and OCP-Zol-1 reduced PVC in experimental groups to 5.2 ± 1.2%, 8.2 ± 1.5%, 5.3 ± 0.7%, 18.2 ± 6.9%, and 62.7 ± 4.7%, respectively (Figure 14C). These data show that 5 days of pre-soaking do not affect the cytostatic properties of the functionalized ceramics (Figure 14B).

However, the viability of tumor cells cultured in the presence of functionalized ceramics increased when the pre-soaking time was extended to 12 and 19 days. Moreover, when functionalized OCP was pre-soaked in CGM for 0 to 12 days, its cytostatic effect decreased with increasing Zol content (Figure 14C). After 19 days of pre-soaking in CGM, this correlation changed: OCP-Cis/Zol-1:0.5 and OCP-Cis/Zol-1:1. In contrast, this caused the most pronounced tumor cell death. The PVC in the experimental groups was 73.2 ± 2.0% and 74.0 ± 3.1%, respectively.

The obtained results confirm the modulating effect of Zol on the OCP functionalization with Cis. Indeed, when the functionalized OCP was pre-soaked in CGM for a long time, the cytostatic effect of OCP-Cis/Zol-1:0.5 and OCP-Cis/Zol-1:1 was greater than that of OCP-Cis-1. This proves that Zol prolongs the cytostatic action of functionalization with Cis ceramics. At the same time, OCP-Cis/Zol-1:0.5 had a cytostatic effect on tumor cells in all periods of observation and had maximum (compared to other samples) effectiveness after 19 days of pre-soaking in CGM. In addition, as shown earlier, OCP ceramics with the same content of Zol had a greater osteoinductive- and osteoclast-inhibiting effect compared to drug-free material.

### 2.9. Biocompatibility Study of Functionalized OCP Ceramics

The study was performed via the subcutaneous implantation of OCP, OCP-Cis, OCP-Zol, and OCP-Cis-Zol granules to mice (four groups of animals). The animals were sacrificed 4, 8, and 12 weeks after surgery, and a histological assessment of the implantation zone and surrounding tissues was performed.

At 4 weeks after the implantation of drug-free OCP ceramics, the implants were surrounded by a multicellular fibrous capsule infiltrated with a small number of lymphocytes. Multinucleated foreign-body giant cells were adjacent to the granules (Figure 15A and Appendix A). After 8 weeks, the growth of the connective tissue between the granules and inside the granules was observed. Numerous blood vessels and foreign-body giant cells adjacent to the granules were visualized in the capsule (Figure 15B and Appendix A). After 12 weeks, the degradation of granules and their gradual replacement with connective tissue intensified (Figure 15C and Appendix A).

Thus, 4–8 weeks after the implantation of OCP-Cis ceramics, histological analysis showed a slight lymphocytic infiltration in the fibrous capsule surrounding the granules, foreign-body giant cells adjacent to the granules, and the intergranular connective tissue ingrowth (Figure 15A,B and Appendix A). Further, 12 weeks after OCP-Cis implantation, no lymphocytic infiltration was observed in the fibrous capsule, and the number of foreign-body giant cells increased (Figure 15C and Appendix A).

Subcutaneous implantation of OCP-Zol caused a completely different response in the tissue. At 4 weeks after implantation, there was a pronounced inflammatory reaction in the connective tissue capsule manifested as lymphoid infiltration. Inside the capsule, fragments of granules were visualized, with an amorphous colloidal substance in the space between them (Figure 15A and Appendix A). After 8 and 12 weeks, the picture in the area of implantation was similar (Figure 15B,C and Appendix A).

At 4 weeks after subcutaneous implantation of OCP-Cis/Zol, the newly formed fibrous capsule was infiltrated by lymphocytes. Inside the implant, an amorphous colloidal substance was observed between the granules (Figure 15A and Appendix A). After 8–12 weeks, the inflammatory reaction subsided, and lymphoid infiltration persisted only in some areas of the fibrous capsule, but the amorphous colloidal substance remained between the fragments of granules (Figure 15B,C and Appendix A).

These data confirm the biocompatibility of OCP ceramics. The OCP functionalization with Cis did not cause activation of the inflammatory response in the tissue, whereas the functionalization of Zol caused aseptic inflammation in the implantation zone. This reaction manifested as a local irritating effect and then gradually subsided.

### 2.10. Osteoconductivity Study of Functionalized OCP Ceramics

A study of the osteoplastic properties of functionalized materials was performed on a rat tibia defect model. Histological study of the implantation zone was carried out at 4, 8, 12 weeks, and 6 months after implantation, sequentially sacrificing two animals.

At 4 weeks after implantation of the drug-free OCP granules, the formation of a thin connective tissue capsule around the implant was seen, and the capsule also filled the spaces between the granules (Figure 16A and Appendix A). Around some granules, areas of newly formed bone tissue and islands of hematopoiesis were visualized (Appendix A). After 8 weeks, the amount of newly formed bone tissue increased, and zones of hematopoiesis were formed between the bone trabeculae (Figure 16B and Appendix A). After 12 weeks (Figure 16C and Appendix A) and 6 months (Figure 16D and Appendix A), the granules were immured in a newly formed bone tissue, and between the bone trabeculae, there were zones of bone marrow hematopoiesis.

OCP-Cis caused the retardation of osteogenesis in the defect zone. At 4 weeks after surgery, the granules were surrounded by a multilayered fibrous capsule growing between them. In the connective tissue adjacent to the surface of the granules, foreign-body giant cells were observed; the newly formed bone tissue was not visualized (Figure 16A and Appendix A). After 8 weeks, islands of newly formed bone tissue were observed in some areas (Figure 16B and Appendix A). The remaining zones between the granules were filled with well-vascularized connective tissue (Appendix A). After 12 weeks, granules surrounded by connective tissue remained in the implant (Figure 16C and Appendix A). They were located closer to the periosteum. In the deeper zone of the defect, the granules were surrounded by rims of newly formed bone tissue, between which there were zones of bone marrow hematopoiesis (Appendix A). At 6 months after surgery, the picture did not differ from that for drug-free OCP: OCP-Cis granules were immured in the newly formed bone tissue, and between the bone trabeculae, there were zones of bone marrow hematopoiesis (Figure 16D and Appendix A).

OCP-Zol caused a pronounced inflammatory reaction in the form of a local irritant effect after implantation. Thus, around the granules, the formation of a shaft of connective tissue, infiltrated with leukocytes, was observed; part of the space inside the granules was filled with an amorphous colloidal substance (Figure 16A and Appendix A). After 8 weeks, inflammatory processes decreased, osteogenesis was activated, bone rod trabeculae were formed around the granules, and bone marrow hematopoiesis was visualized (Figure 16B and Appendix A). After 12 weeks and 6 months, the histological patterns of the defect area were similar. The granules were surrounded by newly formed bone tissue, and between the bone trabeculae, bone marrow hematopoiesis was observed (Figure 16C,D and Appendix A).

Implantation of OCP-Cis/Zol ceramics caused a less pronounced inflammatory response than OCP-Zol. The histological picture, in general, was similar to that after OCP-Zol implantation. At 4 weeks after surgery, a fibrous capsule infiltrated by leukocytes was formed around the implant (Figure 16A and Appendix A). The granules of the material partially degraded and were replaced by an amorphous colloidal substance. Signs of new bone formation appeared after 8 weeks, when the inflammatory process became less pronounced (Figure 16B and Appendix A). After 12 weeks, the amount of newly formed bone tissue increased, and zones of bone marrow hematopoiesis were detected (Figure 16C and Appendix A). After 6 months, the granules were surrounded by bone tissue and, in general, the picture was similar to that observed at 12 weeks (Figure 16D and Appendix A).

### 2.11. Study of Antitumor Properties of Functionalized OCP Ceramics

The antitumor properties of the functionalized materials were studied using a subcutaneous model of a murine mammary Ca-755 adenocarcinoma strain in female C_57_Bl_6_ mice. The control group of tumor-inoculated animals was not subjected to additional manipulations. The animals in the second group received an intravenous injection of Cis simultaneously with tumor inoculation (Cis-i/v). The animals in the two remaining groups, simultaneously with tumor inoculation, were implanted with OCP-Cis into the same axillary region and with OCP-Cis/Zol. A comparative analysis of animals’ survival was carried out using the Kaplan–Meier method (Figure 17A).

The mean life span (MLS) of animals was 22.7 ± 1.2 days in the control group; 38.7 ± 5.3 days—in the Cis-i/v group; 29.7 ± 1.6 days in the OCP-Cis group; and 27.8 ± 3.6 days in the OCP-Cis/Zol group (Figure 17B). In the OCP-Cis group, one of seven animals had complete remission of the tumor nidus: the tumor volume increased up to day 9 and then it began to decrease until complete disappearance by day 14. This animal was alive until the end of the study (90 days) without tumor recurrence. This case was not included when calculating a mean life span.

A comparative analysis showed that, at all time points of the experiment, tumor growth inhibition (TGI) in the OCP-Cis group exceeded that in the Cis-i/v and OCP-Cis/Zol groups, reaching a maximum value of 69.7% on day 25 of the experiment. In the Cis-i/v group, the TGI value did not exceed 27.7% (on day 18); further, it decreased and reached “negative values”, which means a higher tumor growth rate compared to the control group of animals. In the OCP-Cis/Zol group, TGI had an intermediate value (less than in the OCP-Cis group but more than in the Cis-i/v group) at all time points of the experiment (Figure 17C).

The calculation of the effective dose of cisplatin during implantation of OCP-Cis and OCP-Cis/Zol was performed according to the following formula:D = m_ocp_ × N × k/M_mouse_(1)
where D is the effective dose of Cis, m_ocp_ is the mass of functionalized OCP ceramics implanted subcutaneously, N is the quantity of Cis incorporated in 1 mg of OCP, coefficient k is the proportion of Cis released from the material over 5 weeks, and M_mouse_ is the average weight of animals.

For OCP-Cis, the “D” value was 8.34 mg/kg, which was more than 2-times higher than the dose of intravenously administered cisplatin in the Cis-i/v group (4.0 mg/kg). For OCP-Cis/Zol, the effective dose “D” was 4.1 mg/kg, i.e., it was close to the intravenously administered dose of the drug. At the same time, the antitumor effect in this group was more pronounced, as evidenced by TGI.

## 3. Discussion

The limited availability of bone tissue for systemic drug therapy remains a problem for the treatment of a wide range of pathological processes in bones, including oncological. When doses of antitumor drugs are escalated, a therapeutic effect grows much more slowly than serious side effects [8,9,10]. The problem of reduced bone tissue regeneration after surgical interventions in cancer patients is not less serious, requiring the use of osteoplastic materials to fill bone defects after tumor removal.

Hence, the development of a target delivery system for chemotherapy drugs to the bone, using osteoplastic materials, can solve the problem of drug bioavailability, as well as the problem of bone tissue regeneration.

The aim of the work was the development of osteoplastic material with antitumor properties capable of preventing tumor growth in the implantation zone for the replacement of bone defects in oncosurgery. Porous OCP ceramic granules with a relatively high bioresorption rate [5], osteoinductive properties [31,32,33], and high adsorption capacity [38,67] have been used as a platform for functionalization.

It was shown that OCP ceramics are a complex-structured material: two-layered chemical structure of OCP and high porosity of the ceramic OCP granules provide the extended surface area (SSA = 29.9 ± 0.9 mg/m^2^) and high resorption rate. With this, OCP ceramics possess pronounced adsorption properties, as demonstrated for bovine serum albumin and human platelet lysate peptides.

In in vitro experiments, OCP ceramics slowed down the proliferation of osteoblast-like cells at the initial stages of cell growth. This occurred simultaneously with a decrease in calcium ion content and an increase in phosphate ion content in the growth medium. In experiments to simulate the conditions of the internal environment of the body using SBF, it was confirmed that changes in the composition of the medium were caused by the processes of partial hydrolysis of OCP to HA and the precipitation of an amorphous CaP layer on its surface [41,59]. The demonstrated ion exchange processes on the ceramic surface inhibited the proliferative activity of MG-63 cells at the initial stages of cell growth. As a result of the partial hydrolysis of OCP, the ceramic surface became more attractive for cells, influencing their proliferative activity on days 10–14 of cell growth.

With respect to the progenitors of osteoblasts (human BM MSCs) and osteoclasts (mouse macrophages RAW 264.7), OCP ceramics showed an osteoinductive effect, increasing the expression of its marker genes (*RUNX2* and *SP7* in BM MSCs and *TRAP* in RAW 264.7). Moreover, OCP potentiated the inductive action of RANKL on the RAW 264.7 differentiation, and the influence of OCP on the expression of *RUNX2* and *SP7* in BM MSCs was more pronounced than the influence of the osteogenic medium. These data are consistent with numerous studies showing the effect of OCP on the osteogenic differentiation of bone cells [35,36,68,69,70].

In vivo experiments demonstrated the biocompatibility of OCP ceramics and the induction of osteogenesis upon its implantation into a rat tibia defect. However, the rate of new bone formation under the influence of OCP was higher than the rate of ceramic degradation, leading to the immurement of its remnants into the thickness of the newly formed bone. At the same time, the pronounced adsorption capacity of OCP ceramics and cyto- and biocompatibility justified their use as a depot of bioactive compounds and a platform for their delivery to the bone. The chemical composition and structural features of OCP as well as adsorption properties with respect to bioactive compounds determined its osteoinductive effect upon implantation into the tibia defect [41,68,69,71,72,73]. Changes in the ionic composition of the environment induced the differentiation of bone tissue cells, which was additionally activated by blood serum growth factors adsorbed on the surface of the material during implantation into the bone defect. The porosity ensured the ingrowth of blood vessels into the implant, the influx of additional stimulating factors, and progenitor cells that form new bone tissue.

The OCP functionalization with Cis was carried out in an aqueous solution of the drug. As is known, Cis is hydrolyzed in aqueous solution, and its adsorption on the ceramic surface occurs due to electrostatic interaction with negatively charged phosphate groups of OCP [19]. With an increase in the incubation period, the amount of adsorbed Cis decreased, and with an increase in the concentration of Cis to 4 mg/mL, it increased. Obviously, under the chosen conditions, saturation of the OCP surface with the cytostatic was not achieved. Since electrostatic interactions have a weak binding force, the release of Cis from OCP occurred at a high speed. At the beginning of the observation, there was a burst release phase (about one-third of the cytostatic was released in 1 h and more than 70% in the first day), followed by a slower release, which almost completely stopped after 7 days of the study.

Zol was used as a mediator of Cis adsorption and release from the OCP surface. It is known that BPs are inhibitors of bone resorption and are used in the treatment of diseases, such as Paget’s disease, osteoporosis, hypercalcemia, multiple myeloma, osteolytic metastases, etc. [47]. Moreover, BPs’ chemical affinity for calcium ions of calcium phosphates, including HA of bone tissue, is the main feature causing the high binding strength of BPs to bone mineral [11]. This BP capability allows for their use in the local delivery of therapeutic agents to bone tissue through the ability to act as a linker between the drug and the bone mineral. Such systems are the most common components of different compositions of nanoparticles, which are intended for systemic administration into the patient’s body [74,75,76,77]. Applying BPs in combination with platinum agents to obtain dual-targeting systems upon the formation of platinum–BP complexes was also reported [78,79,80].

It was found that, as a result of the functionalization of OCP ceramics with Zol solutions with a concentration of 0.5 to 4.0 mg/mL, the surface of the material was not saturated with the drug. The interaction of Zol with OCP leads to the formation of the second phase, with rod-shaped particles on the ceramic surface. The content of the rods increased with Zol concentration in incorporative solution. Similar elongated particles were revealed by Roussière H. et al. in [61] while treating β-TCP in aqueous Zol solutions. The authors established the formation of a metastable crystalline Zol complex with sodium and calcium, resulting from a partial dissolution of β-TCP. The latter phase transformed into a calcium Zol complex upon repeated washings with water. This suggests that the rods obtained on the surface of the OCP ceramics are the crystalline calcium Zol complex. The XRD patten of the obtained rods on OCP-Zol mostly corresponded to the calcium salt of zoledronic acid in relying on related sources [61,81]. It would be expected that this compound dissolves in an aqueous environment, resulting in a slight transformation or the formation of a Zol complex with the accompanying ions in the solution.

Studying the effect of Zol on the OCP ceramic degradation processes in vitro showed that, during exposure to the SBF model solution, the incorporation of Zol on the OCP surface decreased the intensity of ion exchange processes at the material–solution interface and, as a consequence, the hydrolysis of OCP to HA. The presence of Zol leads to a change in the morphology of the ceramic surface due to the slight OCP degradation and a new phase precipitation by way of binding zoledronate with accompanying cations in the SBF solution. During these processes, an increase in the total mass occurred, especially pronounced at long incubation periods (56 days). Inhibition of the processes of dissolution and hydrolysis of the ceramics in SBF indicates the chemical interaction of Zol with OCP. This fact is confirmed by a slight release of Zol (up to 10%) from the functionalized material over 35 days of observation.

It was found that the functionalization with Zol leads to a change in the bioactive properties of OCP. The inductive effect of ceramics on the differentiation of RAW 264.7 macrophages into osteoclasts was blocked, most pronouncedly by the material with a low content of BP. Indeed, the study of the Zol effect on RAW 264.7 macrophages demonstrated its cytotoxicity at high concentrations (IC50—9.74 µg/mL); thus, only non-toxic concentrations of Zol in OCP blocked the differentiation of RAW 264.7 into osteoclasts. These data correlate with studies that show two possible ways by which bisphosphonates influence osteoclasts: by activating apoptotic cell death and by inhibiting the differentiation of their precursors [11,47,50,82,83]. On the other hand, OCP-Zol enhanced the osteoinductive properties of OCP, as confirmed by the expression of the *ALPL* and *RUNX2* genes in BM MSCs. This is consistent with studies, which also show an increase in osteogenic expression in MG-63 and ADSC cells (*RUNX2*, *ALPL*, collagenase type 1, osteocalcin genes) when they are cultured on alendronate-loaded CaP scaffolds [54,84,85].

At the same time, we showed that the addition of Zol to the incorporative solution affects the binding of Cis to OCP and its release from the material. It was found that an increase in Zol content in the initial solution to 1.0 mg/mL reduced the quantity of Cis incorporated in OCP ceramics, slowing down Cis release at the same time. Cis was completely released from OCP in 5 weeks in the study, while the addition of Zol resulted in the release of 49.1% of Cis (less than half) during the same period. Moreover, the higher the content of Zol in OCP, the slower the release of the cytostatic. The burst release phase was also less pronounced. The duration of Cis release from the obtained functionalized ceramics is significantly higher than the duration of drug release from the materials in similar studies. For example, the release of carboplatin from granulated CaP ceramics in [21] occurred within 5 h. The functionalization of composite ceramic–polymer scaffolds with cisplatin in [44] was more effective: in a release study, 32.7% of the incorporated Cis remained in the material after 10 days, and there was no initial burst release phase. In [86], the complete release of the antitumor drug 5-fluorouracil (5-FU) from three-dimensional polymer-coated CaP scaffolds occurred in 2 h. In [87], the release of an antibiotic from a polymer-coated HA scaffold occurred within 72 h. In [88], the release of doxorubicin from polymeric three-dimensional scaffolds was observed within 14 days.

Thus, it was found that Zol changes the physicochemical and bioactive properties of OCP ceramics, affects the efficiency of its functionalization with Cis, and slows down the kinetics of Cis release from the ceramics. In further in vitro and in vivo studies, we examined the cytostatic and antitumor properties of functionalized OCP ceramics.

Experiments on the human breast cancer cell line MCF-7 showed that OCP ceramics functionalized with Cis and its combination with Zol had a cytostatic effect. Ceramics containing only Zol did not have a pronounced inhibitory effect on MCF-7 cells, which is inconsistent with published studies on the proapoptotic effect of BP on tumor cells [83,89,90]. However, the lack of an effect was most likely due to the low rate of BP release from ceramics.

At the same time, ceramics functionalized with Cis in combination with Zol had the longest duration of action (with initial concentrations in the incorporative solution of 1.0 and 0.5 mg/mL, respectively). The impact of these ceramics on the tumor cell culture for 7 days led to the death of almost the entire cell population (the PVC was 8.3 ± 2.3% from the control). After keeping the functionalized ceramics in the CGM for 19 days, it still caused the death of tumor cells (the PVC was 73.2 ± 2.0% from the control). These data confirm the cytostatic properties of functionalized OCP ceramics and prove their prolonged action.

Next, in vivo studies of biocompatibility and the osteoplastic properties of OCP ceramics functionalized with Cis and Zol were performed. Considering the data obtained in vitro, an aqueous solution of Cis and Zol at concentrations of 1.0 and 0.5 mg/mL, respectively, was used to functionalize OCP.

In a subcutaneous test on mice, the biocompatibility of OCP ceramics was confirmed: a well-vascularized connective tissue without signs of inflammation was formed around the granules; dynamic observation up to 12 weeks showed signs of OCP granule biodegradation. OCP-Cis generally had similar biocompatibility. At the same time, OCP-Zol caused an inflammatory reaction (in the form of a local irritant effect) in the early stages after subcutaneous implantation. In addition, a weakly degrading amorphous colloidal substance was formed in the zone of OCP-Zol granule implantation in the bone defect. A similar aseptic inflammatory reaction was also observed after subcutaneous implantation of OCP-Cis/Zol.

The osteoplastic properties of the obtained OCP ceramics were studied on a rat tibia defect model. Drug-free OCP ceramics confirmed the osteoinductive properties: as early as 4 weeks after implantation, cancellous bone with foci of hematopoiesis was formed around the granules, and after 12 weeks, the reorganization of the newly formed bone tissue in the defect zone was completed. OCP-Cis slowed down osteogenesis: areas of newly formed bone tissue were formed around the granules only 8 weeks after implantation. Implantation of both OCP-Zol and OCP-Cis/Zol into a tibia defect, as in the case of subcutaneous injection, caused an aseptic inflammatory reaction and the formation of an amorphous colloidal substance in the zone of granule implantation. After 8 weeks, the inflammatory process began to subside, neo-osteogenesis was activated, and after 6 months, the reconstruction of the bone defect with the newly formed bone tissue was completed. However, similar studies of BP-loaded bone substitutes did not describe such an inflammatory reaction. On the contrary, these studies demonstrated the stimulation of neoosteogenesis by BP-loaded materials and a significant increase in bone mass in the defect during their implantation compared to the controls [54,85,91,92]. The amorphous colloidal substance found in the OCP-Zol granules, both in subcutaneous implantation and in the tibia defect, is most likely the result of the interaction between Zol and OCP. These data agree with the results of a physicochemical study of OCP-Zol and in vitro experiments, performed through exposure of OCP-Zol to the model solution, SBF. The formation of a soluble Zol complex on the OCP-Zol surface was revealed in the form of rod-shaped particles, which were transformed into the further Zol phase as a result of dissolving and binding with cations of SBF. Probably, the amorphous colloidal substance in vivo is similar to the Zol complex with cations of the solution in vitro.

In a subcutaneous model of Ca-755 strain in mice, it was shown that simultaneous implantation of the studied materials in the tumor area resulted in an increase in the MLS and TGI compared to control (untreated animals). The increase in these parameters in the group receiving OCP-Cis was more pronounced than in the group receiving OCP-Cis/Zol. This is most likely due to the lesser quantity of Cis bound to OCP upon combined functionalization with Zol. At the same time, it should be noted that the antitumor effect in the OCP-Cis/Zol group exceeded the effect in the Cis-i/v group, with equal effective doses of cytostatic in these groups. In addition, according to the data presented above, during the study period (35 days), about half of the incorporated Cis was released from OCP-Cis/Zol, while all of the cytostatic was released from OCP-Cis.

In clinical practice, to prevent continued tumor growth after removal of the primary lesion, it is necessary to create a constant and long-lasting cytostatic concentration in the defect zone. Therefore, the implantation of OCP ceramics functionalized with Cis in combination with Zol, which had the longest Cis release period and cytostatic effect in vitro, is more likely capable of providing a longer-term prevention of tumor recurrence.

## 4. Materials and Methods

### 4.1. Chemicals

The following reagents of high chemical purity were used for the synthesis of powders and ceramic technology: calcium nitrate tetrahydrate Ca(NO_3_)_2_·4H_2_O, ammonium hydrophosphate (NH_4_)_2_HPO_4_, sodium acetate CH_3_COONa, phosphoric acid H_3_PO_4_, and an aqueous solution of ammonia NH_4_OH.

The following reagents of high chemical purity were used to prepare the model SBF solution: sodium chloride NaCl, sodium hydrogencarbonate NaHCO_3_, potassium chloride KCl, potassium hydrogen phosphate trihydrate K_2_HPO_4_·3H_2_O, magnesium chloride hexahydrate MgCl_2_·6H_2_O, calcium chloride dihydrate CaCl_2_·2H_2_O, and sodium sulfate Na_2_SO_4_.

For the functionalization of ceramics, the following drugs were used: cisplatin (*cis*-Diaminedichloroplatinum, Cl_2_H_6_N_2_Pt) (Cat. No. 15663-27-1) and zoledronic acid ([1-Hydroxy-2(1H-imidazole-1-yl)ethylidene]bis[phosphonic acid], C_5_H_10_N_2_O_7_P_2_) (Cat. No. 165800-08-61) of 99.9% purity from Sigma-Aldrich (St. Louis, MO, USA).

### 4.2. Fabrication of OCP Ceramics

Porous OCP granules were obtained through tricalcium phosphate (TCP) powder preparation step followed by TCP-based ceramics fabrication and chemical transformation of TCP to OCP [59,93].

TCP powder was prepared via precipitation technique from aqueous solutions of Ca(NO_3_)_2_·4H_2_O and (NH_4_)_2_HPO_4_ to achieve the ratio Ca/P = 1.5; pH value was maintained at 7.0 by addition of ammonia solution. The precipitate was filtered and washed with distilled water, then dried at 70 °C. TCP powder was obtained by grinding the dried precipitate and sieving through a mesh with a cell size of 0.1 mm.

TCP ceramics were fabricated via replica method using impregnation of polyurethane foams with the TCP powder water slurry. The impregnated foams were dried at 70 °C and then slowly sintered at 1350 °C to obtain the ceramic foam replica. The resulting ceramics were carefully crushed to pieces, then the fraction of 0.5–1.0 mm was selected by sieving between meshes. The TCP ceramic granules were approved to be monoclinic tricalcium phosphate phase (α-TCP) via X-ray diffraction (XRD) and Fourier-transform infrared spectroscopy (FTIR).

Ceramic TCP granules were chemically transformed to the OCP phase saving macrostructure features according to a previously reported method [59]. Sequential conversion of TCP into dicalcium phosphate dihydrate (DCPD) and DCPD into OCP was performed by aging granules in an acetate buffer in glass vessel at a ratio of material/solution (1:100) at constant slow stirring using thermostatic shaker incubator (BioSan, Riga, Latvia) [94]. Briefly, the chemical transformation of α-TCP to DCPD was performed in an acetate solution in the presence of phosphoric acid over 7 days, while maintaining 37 °C and pH level 5.5 ± 0.1. The obtained DCPD granules were examined using XRD and FTIR. Then, DCPD was hydrolyzed in 1.5 M sodium acetate solution at 37 °C and pH level 9.0 ± 0.2 over 7 days. The resultant OCP granules were carefully washed several times with distilled water to achieve the constant pH of washing solution of 7.0 ± 0.5, then dried at 50 °C, and sieved through a mesh with a cell size of 0.1 mm.

### 4.3. Material Characterization

The fabricated materials were characterized using XRD and FTIR at various stages of technology and investigation. XRD patterns of the samples preliminarily ground into powder were obtained using Shimadzu XRD-6000 diffractometer (Kyoto, Japan) with Cu Kα radiation (λ = 1.5418 Å), 40 kV, and 200 mA. The angular-analyzed interval 2Theta was from 4 to 60°, with a step of 0.02°, and speed of the counter was 2°/min. The phases were identified according to JSPDS. FTIR spectra of the materials were recorded using a Bruker Vertex 70V vacuum IR spectrometer (Billerica, MA, USA) in a range of 400–4000 cm^−1^ with a resolution of 4 cm^−1^ using the ATR device in Transmission mode. The specific surface area (SSA) of the materials was measured using Brunauer–Emmett–Teller (BET) method using the TriStar 3000 Micromeritics analyzer (Atlanta, GA, USA). The microstructure study of the materials was performed with scanning electron microscopy (SEM) using Tescan Vega II (Brno, Czech Republic), and the samples were previously coated with gold using Q150R device, Quorum Technologies (Ashford, England). The element content (calcium (Ca), phosphorus (Pi), platinum (Pt)) was analyzed for ceramic samples previously dissolved in HCl or for samples of incorporative solutions by inductively coupled plasma atomic emission spectroscopy (ICP-AES) using Perkin Elmer Optima 5300DV spectrometer (Waltham, MA, USA).

### 4.4. Functionalization of OCP Ceramics with Drugs

The functionalization of OCP ceramics with drugs (Cis, Zol, and their combination) was carried out by exposing OCP granules to aqueous solutions of drugs with various concentrations (0.5–4.0 mg/mL) for various periods (from 1 to 7 days). As a result of OCP granule exposure to drug solutions, their incorporation on OCP surface took place; the effectiveness of incorporation depended on the type of drug and experimental conditions. The aqueous solutions of drugs for functionalization were named incorporative or initial solutions.

The functionalization of OCP ceramics was carried out using an aqueous solution of Cis with a concentration of 0.5; 1.0; 2.0; 4.0 mg/mL, an aqueous solution of Zol with a concentration of 0.2; 0.5; 1.0; 2.0; 4.0 mg/mL, and an aqueous solution, which was a combination of Cis and Zol (with Cis concentration of 1.0 mg/mL and Zol concentration of 0.2, 0.5, and 1.0 mg/mL).

To functionalize the ceramics with drugs, the OCP granules (50.2 ± 0.02 mg) were placed in plastic tubes containing 1 mL of the drug solution; next, the tubes were placed in a thermostat (37 °C) for 48 h with constant stirring at 150 rpm (orbital shaker Elmi Shaker ST-3, Riga, Latvia). After 48 h of incorporation, the drug solution was carefully decanted, and the OCP granules were washed once with distilled water and dried at 37 °C for 24 h.

OCP ceramics functionalized with Cis were designated as OCP-Cis, OCP ceramics functionalized with Zol were designated as OCP-Zol, and functionalized OCP ceramics obtained using an aqueous solution of Zol at concentrations of 0.2, 0.5, and 1.0 mg/mL were designated as OCP-Zol-0.2, OCP-Zol-0.5, and OCP-Zol-1, respectively. Functionalized OCP ceramics obtained using a combination of Cis and Zol solutions at concentrations (mg/mL) of 1:0.2, 1:0.5, and 1:1 were designated as OCP-Cis/Zol-1:0.2, OCP-Cis/Zol-1:0.5, and OCP-Cis/Zol-1:1, respectively.

### 4.5. In Vitro Study of Functionalized OCP Ceramic Biodegradation

In vitro study of the biodegradation of drug-free and functionalized ceramics was performed by exposing OCP granules to a model SBF solution for 1, 3, 7, 14, 28, and 56 days. The concentration of Zol solution used for the functionalization was 0.5 and 1.0 mg/mL.

SBF solution was prepared in accordance with the classical methods [57,95]. The following high-purity reagents were sequentially dissolved in distilled water preheated to 37 °C: NaCl, NaHCO_3_, KCl, K_2_HPO_4_·3H_2_O, MgCl_2_·6H_2_O, CaCl_2_·2H_2_O, and Na_2_SO_4_. The pH value was maintained at 7.2–7.4 with an addition of tris(hydroxymethyl)aminomethane and HCl.

For the procedure, the OCP granules (100.0 ± 0.2 mg) were placed into plastic tubes with SBF in a ratio of material/solution (1:60). The soaking was performed in shaker-incubator at 37 °C (BioSan, Riga, Latvia). At each observation time, the samples were taken out and separated by decanting. The granules were carefully washed with distilled water and dried at 40 °C for a day. The samples of the SBF after the experiment were analyzed for the content of calcium (Ca^2+^) and phosphor (Pi). The granules were examined to study the loss/gain of mass, microstructure, and FTIR spectra.

### 4.6. Drug Release Study

The functionalized OCP ceramics were placed in tubes with a Dulbecco’s phosphate-buffered saline (DPBS) (the buffer volume was determined based on a ratio of 1 mL DPBS per 50 mg of material). At the specified time points (0.5, 1, 4, 24, 72 h, 1, 2, 3, 4, 5 weeks), the entire buffer solution was collected (950 µL) to assess the content of the studied drug in it, after which the same volume of fresh buffer was added to the material (950 µL). The concentration of the drugs in the obtained samples was determined. The total quantity of released drug was calculated as the sum of its content in the samples at each time point. The results are presented in recalculation for 1 mg of functionalized material.

### 4.7. Determination of the Drug Content in Functionalized OCP Ceramics

The quantity of the drug incorporated in 1 mg of OCP was assessed by determining the concentration of the drug in the incorporative solution before and after OCP ceramics were exposed to it, as well as by determining the concentration of the drug in the functionalized OCP ceramics dissolved in 1H HCl.

### 4.8. Determination of the Drug Concentration in Solution

Cisplatin concentration in solutions was determined by quantifying the Pt content via ICP-AES using ICE 3000 spectrometer (Thermo Fisher Scientific, Waltham, MA, USA) in an air–acetylene flame with a deuterium background corrector. Lamp with a hollow cathode was used as a resonance radiation source. The platinum analysis conditions were optimized, taking into account the composition of the matrix and analytes, as shown in [96].

The concentration of zoledronic acid in solutions was determined via high-performance liquid chromatography. For this purpose, the Agilent 1260 Infinity (Agilent Technologies, Santa Clara, CA, USA) chromatographic system with a Zorbax Eclipse C18-A column was used. A mixture of methanol and phosphate buffer solution in a ratio of 5:95 with the addition of sodium pyrophosphate (Lenreactive, Moscow, Russia) at a concentration of 1.5 mmol/L was used as the mobile phase. The methodology was based on a method described earlier in the literature [97].

### 4.9. Cell Culture

In vitro experiments were carried out with the use of the following continuous cell lines: osteoblast-like cells of human osteosarcoma MG-63 (CRL-1427^TM^, American Type Cell Collection (ATCC)), human breast cancer (BC) cells MCF-7 (HTB-22^TM^, ATCC), mouse macrophages RAW 264.7 (TIB-71, ATCC), and with the use of donor-derived primary cell culture of human bone marrow mesenchymal stem cells (MSC BM) of 3–8 passages. Obtaining an intraoperative sample of biological material from patient A was performed after voluntary informed consent of the patient and was approved by the Independent Ethics Council of the P.A. Herzen Moscow Cancer Research Institute. In vivo experiments were carried out with the use of the continuous cell line of murine mammary Ca-755 adenocarcinoma.

All manipulations with cell cultures were performed under standard aseptic conditions. The cells were cultured in complete growth medium (CGM) based on Dulbecco’s modified Eagles medium (DMEM) (PanEco, Moscow, Russia) supplemented with 10% fetal calf serum (HyClone, South Logan, UT, USA), 60.0 mg/mL L-glutamine (PanEco, Moscow, Russia), 50.0 µg/mL gentamicin (PanEco, Moscow, Russia), and 20.0 mM Hepes buffer (PanEco, Moscow, Russia) in humid air at 37°C and 5% CO_2_. The medium was changed twice a week. Cell cultures in the preconfluent monolayer were used for experiments.

### 4.10. Assessment of Cell Culture Viability Using the MTT Assay

The MTT assay is based on the ability of live cell dehydrogenases to reduce 3-(-4,5-dimethylthiazolyl-2)-2,5-diphenyltetrazolium bromide (MTT, Sigma-Aldrich, St. Louis, MO, USA) into water-insoluble blue formazan crystals [98].

To perform the MTT test, an MTT solution at a concentration of 5.0 mg/mL was added to the cell culture medium. After 3 h of incubation (5% CO_2_, 37 °C), the formed formazan was dissolved with isopropyl alcohol, and the remains of lysed cells were precipitated via centrifugation (10 min at 3000 rpm). In the resulting supernatant, the optical density (OD) of the formazan solution was evaluated on a Multiscan FC spectrophotometer (Thermo Scientific, Waltham, MA, USA) at a wavelength of 540 nm. The calculation of the population of viable cells (PVCs) in relation to the control (in %) was carried out according to Formula (2):PVC = OD_exp_/OD_contr_ × 100 (%),(2)
where OD is the value of the optical density of the formazan solution.

The control was non-treated cell culture, incubated under standard conditions.

### 4.11. Evaluation of OCP Ceramic Cytotoxicity

OCP ceramics were sterilized at 120 °C for 2 h using thermostat (Binder, Tuttlingen, Germany). The physicochemical study of the sterilized OCP ceramics in comparison with non-treated ceramics is presented in Appendix A. To prepare the extraction solution, 0.22 g of sterile granular ceramics was placed in 1.2 mL of sterile CGM, incubated for 24 h at 37 °C with constant stirring, and centrifuged at 2000 rpm for 10 min. MG-63 cells with a seeding density of 15.0 × 10^3^ cells/cm^2^ were cultured until a subconfluent monolayer was reached with the following complete replacement of CGM with ceramic extracts. After 24 h, the viability of the cell culture was analyzed using the MTT assay [99].

### 4.12. Evaluation of OCP Ceramic Cytocompatibility

Sterile OCP ceramics (120 °C, 2 h, thermostat (Binder, Tuttlingen, Germany) were placed into the wells of 96-well plates at 8.3 mg/well, and a suspension of MG-63 cells was introduced with a seeding density of 15.0 × 10^3^ cells/cm^2^. Culturing was carried out for 1, 3, 7, 10, and 14 days, and medium was changed every 3 days. Next, the viability of the cell culture was analyzed using the MTT assay [99].

### 4.13. Assessment of Cell Culture Sensitivity to the Drugs

The sensitivity of MCF-7 and RAW 264.7 cell lines to the cisplatin and zoledronic acid was studied in a concentration range of 0.001–100 µg/mL with a cell seeding density of 15.2 × 10^3^ cells/cm^2^ and 10.02 × 10^3^ cells/cm^2^, respectively. To assess the toxicity of drugs, the IC_50_ value was calculated as a concentration of the studied drug that caused the death of 50% of the cell population, compared with the control. Cells cultured on plastic under standard conditions served as controls. Cell culture viability was assessed using the MTT assay.

### 4.14. Assessment of Cytostatic Properties of Functionalized OCP Ceramics

The study was performed on MCF-7 cells using the following models:-The impact of functionalized OCP ceramics on tumor cells in the dynamics of culturing (1, 3, and 7 days);-The impact of functionalized OCP ceramics, preliminarily exposed to the CGM (5, 12, and 19 days), on tumor cells over 7 days.

MCF-7 cells were cultured in 24-well plates with a seeding density of 20 × 10^3^ cells/cm^2^. Sterile samples of OCP ceramics (50 mg each) were placed in the inner chamber of transwell inserts (pore diameter 8 μm) with further placement in wells with cells. The volume of the CGM in each well with inserts was 1 mL. Next, the viability of cell cultures in the experimental wells was assessed using the MTT assay in comparison with the control. Cells cultured on plastic under standard conditions served as controls.

### 4.15. Directed Osteogenic Cell Differentiation

Directed differentiation of RAW 264.7 macrophages into osteoclasts in vitro was performed by adding the RANKL factor at a concentration of 40–80 ng/mL to the culture medium during cell seeding and medium change. The differentiation efficiency was assessed after 5–14 days of culturing cell growth using light microscopy (cells containing 3 or more nuclei were considered osteoclasts) and measuring the expression of the osteoclastic marker gene, tartrate-resistant acid phosphatase (*TRAP*), via real-time PCR.

Directed differentiation of BM MSCs into osteoblasts was performed by culturing in the osteogenic medium StemPro Osteogenesis Differential Kit (Thermo Fisher Scientific, Waltham, MA, USA) for 1–3 weeks using the manufacturer’s standard protocol. The differentiation efficiency was assessed on days 7–21 by measuring the expression of the osteoblastic marker genes *RUNX2*, *SP7*, and *ALPL*, using real-time PCR.

### 4.16. Study of the Expression of Marker Genes

The changes in the expression level of marker genes in cells were assessed using real-time PCR. On the 14th day of cell growth, the following genes were analyzed: *RUNX2*, *SP7*, *ALPL*, and *TRAP*. The expression of mentioned genes was normalized using the glyceraldehyde-3-phosphate dehydrogenase gene (housekeeping gene, *GAPDH*).

The total RNA pool was isolated by phenol-chloroform extraction using Qiazol lysis buffer (Qiagen, Hilden, Germany) followed by purification on RNeasy Mini Kit columns (Qiagen, Hilden, Germany). The amount of RNA in the samples was measured using a NanoDrop ND-2000 spectrophotometer (Thermo Scientific, Waltham, MA, USA). Reverse transcription was performed using the MMLV RT kit (Evrogen, Russia) using Random (dN)10-primer. qPCRmix-HS SYBR with SYBR Green I intercalating dye (Evrogen, Moscow, Russia) and a DTlite detecting amplifier (DNA-Technology, Moscow, Russia) were used for real-time PCR.

The amplification program included a “hot start” (94 °C, 10 min) followed by 50 cycles of template denaturation (94 °C, 20 s), primer annealing (64 °C, 10 s), and amplicon elongation (72 °C, 15 s). All samples were analyzed in triplets.

The expression of the studied genes was analyzed using the ΔΔCt algorithm. With this algorithm, the expression level of studied genes in both the test samples and control samples was adjusted in relation to the expression of normalizer gene (*GAPDH*) from the same two samples. As a result, the fold difference (FD) in the expression level of the studied gene in the test and control groups was calculated. The control was non-treated cell culture, incubated under standard conditions.

Synthesis of oligonucleotide sequences of primers of target and normalization genes was carried out by Evrogen (Moscow, Russia) at our request. The sequences of primers of the analyzed genes are shown in Appendix A.

### 4.17. In Vivo Study

The experiments on small laboratory animals were carried out in compliance with the principles of humanity and the requirements formulated in the Directive of The European Parliament and of the Council 2010/63/EU “On the protection of animals used for scientific purposes” and according to the protocol №2—SI-00010, approved on 4 August 2021 by the Commission for Bioethical Control over the housing and use of laboratory animals for scientific purposes of the National Medical Research Radiological Centre. The animals were housed in a vivarium under standard light, food, and water conditions. The samples of materials used in the experiments were sterilized by γ-irradiation (15 kGy). Immediately before surgery, the samples were soaked in saline. Animals were operated under general anesthesia.

Samples of drug-free and functionalized granular OCP ceramics were included in the in vivo study. The functionalization with Cis and Zol was carried out via adsorption from aqueous solutions containing 1.0 mg/mL of the drugs. Combined functionalization was performed using an aqueous solution with Cis (1.0 mg/mL) and Zol (0.5 mg/mL). The distribution of animals into groups is described in Table 1.

### 4.18. Study of Material Biocompatibility

The biocompatibility of the developed materials was evaluated on a model of subcutaneous implantation in male mice of the BDF1 line weighing 18–20 g (FSBIS Scientific Center for Biomedical Technologies of the Federal Medical-Biological Agency of Russia, Andreevka branch). Samples of sterile materials were placed under the skin of the back at the level of the thoracic spine of mice under general anesthesia (intraperitoneally, 0.1 mL of ketamine/relanium mixture in a ratio of 1:1).

At a certain time point (4, 8, 12 weeks), 2 mice from each group were sacrificed using a lethal dose of carbon dioxide. Samples of materials were removed, visually assessed using a stereomicroscope and a digital video camera (Olympus, Tokyo, Japan), fixed in 10% formalin solution, and placed in paraffin blocks. Histological sections were stained with hematoxylin and eosin and photographed using an Eclipse Ti microscope equipped with a DS-Fi1c digital camera (Nikon, Tokyo, Japan).

### 4.19. Study of Material Osteoconductive Properties

The osteoconductive properties of the materials were studied on the model of tibia defect in male Wistar rats weighing 180–200 g. For this purpose, the animals underwent marginal resection of the tibia. The surgery was performed under general anesthesia (preliminary sedation via intraperitoneal injection with 0.5 mL of 0.25% solution of droperidol, followed by intramuscular injection with 0.25 mL of 0.25% solution of ketamine). Next, the animal was placed in a supine position, and a skin incision of 2–2.5 cm was made along the inner medial surface of the leg exposing the tibia. Next, it was cleaned from the periosteum and “fenestrated” defect of 6–8 mm in length, 1.5–2.0 mm in width, and 1.5–2.0 mm in depth was formed with a bur on the border of the upper and middle thirds of the bone with penetration into the bone canal. In accordance with the study protocol, sterile samples of materials were placed in the area of the defect, and, at the final stage, the muscles and skin of the thigh were sutured in layers.

At a certain time point (4, 8, 12 weeks, 6 months), 2 rats from each group were sacrificed using a lethal dose of carbon dioxide. A bone fragment in the area of the defect and adjacent tissues was removed and fixed in 10% formalin solution. Next, a microCT study was performed on Skyscan 1275 (Bruker, Kontich, Belgium), followed by decalcification with a 0.3 M solution of ethylenediaminetetraacetic acid (EDTA) for 30 days. After histological processing, tissue sections were prepared, stained with hematoxylin and eosin, and the morphology of the bone defect was analyzed using an inverted light microscope Nikon Eclipse Ti (Tokyo, Japan).

### 4.20. Study of Material Antitumor Properties

The study of the antitumor properties of materials was performed on a subcutaneous model of murine mammary Ca-755 adenocarcinoma strain in mice, with an average mass of 26 g. The tumor strain was maintained in C57Bl6 female mice through subcutaneous inoculation of 100 mg of minced tumor tissue suspension in 0.5 mL of Eagle’s medium every 10–14 days. For experiments, female mice were inoculated with 50 mg of tumor tissue suspension diluted in 0.3 mL of Eagle’s medium under the skin (in the area of the mammary glands). Simultaneously with tumor inoculation, sterile samples of drug-free and functionalized materials were implanted into the inoculation zone. An additional group of animals was introduced into the study, receiving a single intravenous injection of Cisplatin-Teva (1.0 mg/mL, Canpol Sp. Z o.o. SKA, Warsaw, Poland) at a dose of 4.0 mg/kg (Cis i/v, group 5) simultaneously with tumor inoculation. Over the next 90 days, the animals were observed every 2 days: the volume of the formed tumors was measured, and alive animals were counted.

The antitumor effect of functionalized materials was assessed by analyzing the survival of animals in groups using the Kaplan–Meier method, comparing the mean life span (MLS) in groups and calculating the degree of tumor growth inhibition (TGI) compared with the control. For this purpose, the volumes of formed tumors in animals in groups were measured at each time point of observation. The TGI value was calculated busing Formula (3):TGI = (Vcontrol − Vexperiment)/(Vcontrol) × 100 (%),(3)
where V is the average tumor volume (mm^3^) in the experimental and control groups, respectively.

### 4.21. Statistical Analysis

Statistical analysis of the obtained data was performed using STATISTICA 10.0 software (Statsoft, Inc., USA, 2011). At least three independent observations were made in each experiment. The normality of the distribution of values in the samples was checked using the Shapiro–Wilk W-test. The significance of differences in intergroup means was assessed using Student’s *t*-test. Differences were considered statistically significant at *p* < 0.05.

## 5. Conclusions

OCP ceramics were studied as a platform for functionalization with cytostatics for targeted delivery to bone tissue. It was shown that OCP has a highly developed surface and porosity, ensuring its good adsorption capacity and cytocompatibility. It was found that OCP ceramics have osteoinductive properties, activating the differentiation of mouse RAW 264.7 macrophages into osteoclasts and human BM MSCs into osteoblasts.

Conditions for the incorporation of Cis into OCP and Cis in combination with BP Zol into OCP were developed. It was shown that Zol slightly reduced the amount of Cis bound to OCP but ensured its smooth and prolonged release from OCP.

Functionalization with Zol changes the physicochemical and bioactive properties of OCP ceramics: the rate of ceramic biodegradation decreases, osteoinductive properties in relation to BM MSCs increase, and, on the contrary, they are blocked in relation to RAW 264.7 macrophages.

OCP ceramics functionalized with Cis and its combination with Zol exert a cytostatic effect on MCF-7 tumor cells; combined functionalization provides OCP ceramics with the longest duration of a cytostatic effect in vitro. Using a subcutaneous implantation model in mice, it was shown that both the original OCP and OCP-Cis ceramics are biocompatible materials. The presence of Zol somewhat reduced the biocompatibility of OCP, which was manifested in a local aseptic inflammatory reaction in the early stages after implantation (up to 8 weeks). OCP ceramics were found to have pronounced osteoconductive and osteoinductive properties, as evidenced in the rat tibia defect model. The presence of Cis and Zol in ceramics slows down the process of osteogenesis in defects in the early stages after implantation (up to 4 weeks), which is further completed with organotypic replacement of the defect with bone tissue. The presence of Zol in the implant causes an aseptic inflammatory reaction in the defect in the early periods (up to 4 weeks), subsiding in later periods.

On the subcutaneous model of Ca-755 strain in mice with simultaneous implantation of functionalized OCP ceramics in the tumor zone, OCP-Cis and OCP-Cis/Zol ceramics were shown to exert a more pronounced inhibitory effect on tumor growth, compared to intravenous administration of Cis.

Therefore, the developed method for the functionalization of OCP ceramics with cisplatin in combination with zoledronic acid provides a material with prolonged cytostatic and antitumor properties. The use of these materials in clinical practice during the surgical removal of primary or metastatic tumors in bone tissue seems promising.

## 6. Strengths and Limitations of This Study

The detailed characterization of the biological properties of drug-free OCP ceramics and OCP ceramics, functionalized with a Zol, Cis, and Cis-Zol combination, was performed.The cytostatic properties of the developed materials were demonstrated in in vitro experiments and confirmed by in vivo studies. As a result, the antitumor effect of the obtained functionalized materials and their advantages over intravenous administration of cytostatics were established.The nature of the interaction between Cis, Zol, and OCP ceramics was described implicitly, based on the literature data and results of XRD analysis, FTIR, SEM, and drug release study. In addition, it was shown that the functionalization with Zol leads to the formation of a crystalline calcium Zol complex on the OCP surface. Zol incorporation due to chemosorption decreases the hydrolysis rate of OCP to HA and increases the total mass of the material. Further, Zol release was found to be slight over 35 days of observation (10% of incorporated amount). Beyond that, it is proved that the formation of this complex increases the binding strength of Cis to OCP and leads to a significant slowdown in its release.

Our subsequent studies will be aimed at identifying the nature of the interaction of zoledronate and cisplatin and their combined incorporation into the OCP.

4.OCP ceramics functionalized with Cis possess a more pronounced antitumor effect than ceramics functionalized with a combination of Cis and Zol. These results are due to the higher content of Cis in OCP when using this type of functionalization. An increase in the initial concentration of Cis in the incorporative solution with combined functionalization will inevitably lead to an increase in its content in the material and, as a consequence, to an increase in the cytostatic and antitumor effects of the material.

## 7. Patents

Kuvshinova E.A, Shanskiy Y.D, Petrakova N.V, Nikitina Yu.O., Sviridova I.K., Komlev V.S., Sergeeva N.S., Kaprin A.D. Method of functionalization of calcium phosphate material by cisplatin preparation in aqueous solution. RU Patent № 2765465 from 31 January 2022.

## Figures and Tables

**Figure 1 ijms-24-11633-f001:**
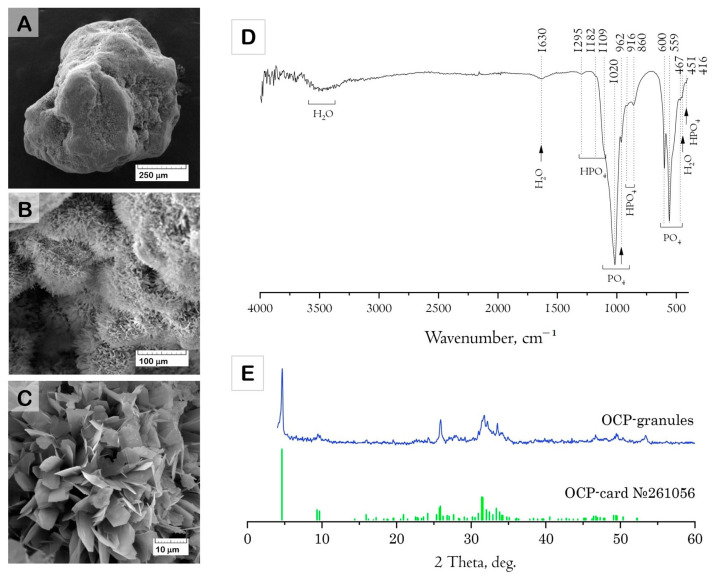
Characterization of initial OCP ceramics intended for functionalization: (**A**–**C**)—SEM images at various magnifications to demonstrate macro- and microstructure; (**D**)—FTIR spectrum; (**E**)—XRD pattern with Bragg peaks positions for OCP (No. 261056).

**Figure 2 ijms-24-11633-f002:**
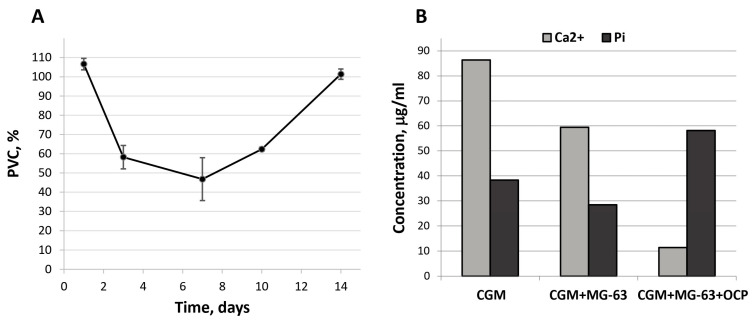
Data of observation in direct contact of MG-63 cells with OCP ceramics. (**A**)—PVC in culture under the influence of OCP in comparison with control (MTT assay, control—untreated MG-63 cells); (**B**)—changes in ionic composition of CGM, (5 days of observation).

**Figure 3 ijms-24-11633-f003:**
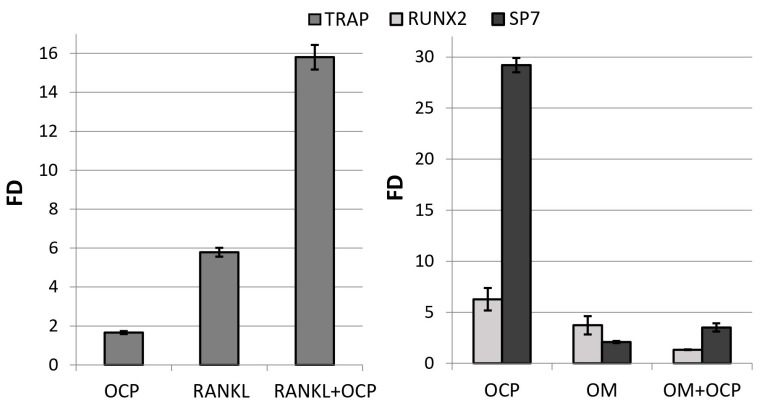
Changes in the *TRAP* expression in RAW 264.7 cells and the *RUNX2* and *SP7* expression in BM MSCs under influence of the OCP, RANKL (40 ng/mL) and osteogenic medium (OM). Control—untreated RAW 264.7 cells/BM MSCs., normalization genes—mouse and human *glyceraldehyde-3-phosphate dehydrogenase* (*GAPDH*).

**Figure 4 ijms-24-11633-f004:**
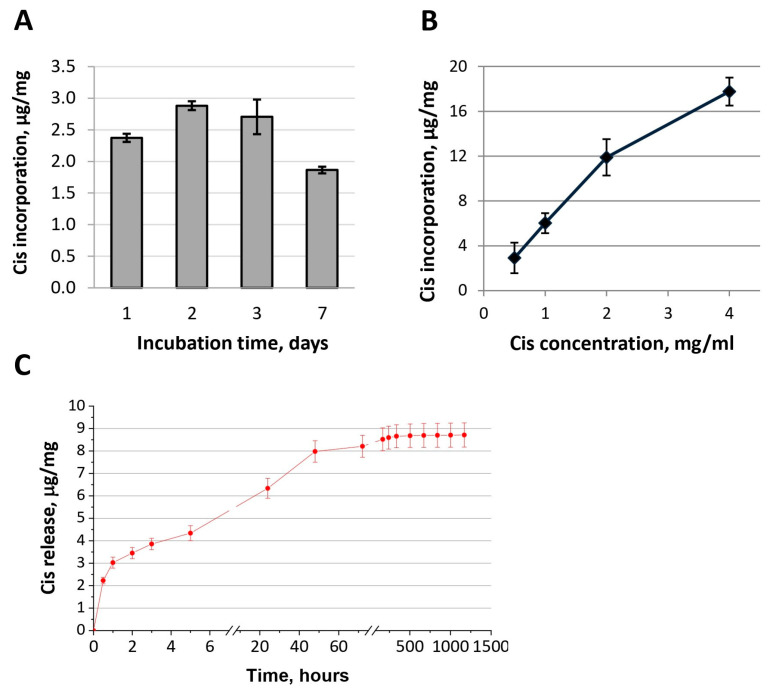
Efficiency of Cis incorporation into OCP ceramics depending on the incubation time (**A**) and Cis concentration in the initial solution (incubation time—2 days) (**B**); kinetics of Cis release from OCP ceramics (incubation time—2 days, initial Cis concentration—1 mg/mL) (**C**).

**Figure 5 ijms-24-11633-f005:**
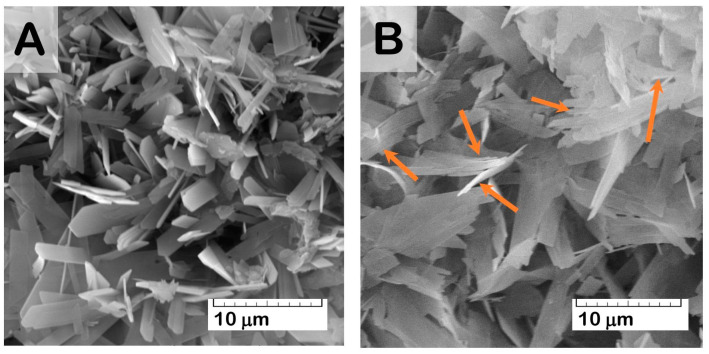
SEM images of the surface of OCP ceramics after incubation in Cis water solution during 1 (**A**) and 7 (**B**) days. Arrows demonstrate the thinning and reducing of OCP crystals with irregular faces during a long-term incubation.

**Figure 6 ijms-24-11633-f006:**
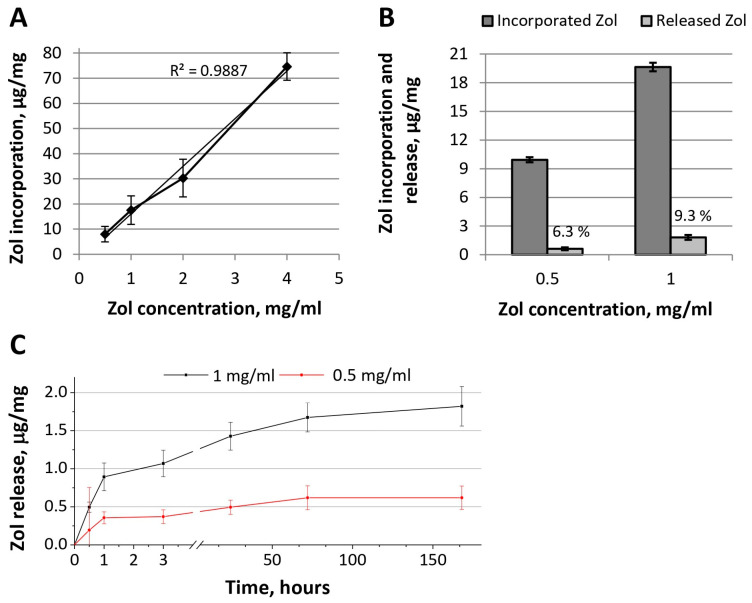
Efficiency of Zol incorporation into OCP ceramics (**A**), the ratio of released Zol to incorporated (**B**), and the kinetics of Zol release from functionalized OCP ceramics (**C**) depending on its concentration in the initial solution. Time of incorporation in drug solutions—2 days; time of observation of Zol release—35 days.

**Figure 7 ijms-24-11633-f007:**
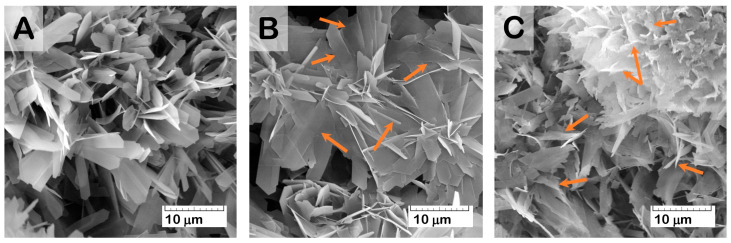
SEM images of OCP ceramics: initial OCP crystals with well-shaped borders (**A**); OCP granules functionalized with Zoledronic acid at concentration of 1 mg/mL (**B**) and 4 mg/mL (**C**). Arrows show the degraded thinned crystals.

**Figure 8 ijms-24-11633-f008:**
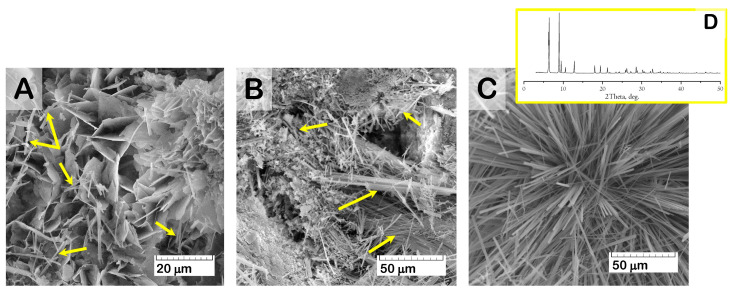
SEM images of OCP-Zol ceramics demonstrate the formation of a new rod-shaped phase due to interaction between Zol and OCP during incorporation. Arrows indicate the rods, the number of which rises with increasing concentration. After exposure to Zol solution of 1 mg/mL (**A**), 2 mg/mL (**B**), 4 mg/mL (**C**), XRD pattern of Zol rods (**D**).

**Figure 9 ijms-24-11633-f009:**
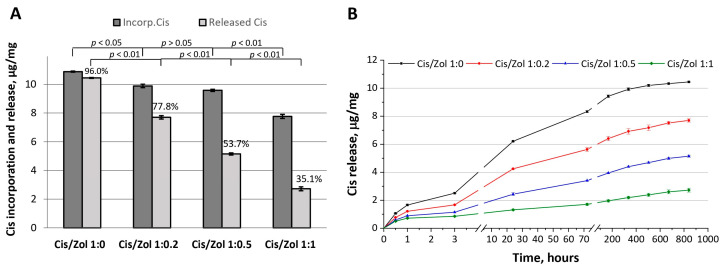
Efficiency of Cis incorporation into OCP ceramics, the ratio of released Cis to incorporated Cis (**A**), and the kinetics of its release (**B**). Time of observation of Cis release—35 days.

**Figure 10 ijms-24-11633-f010:**
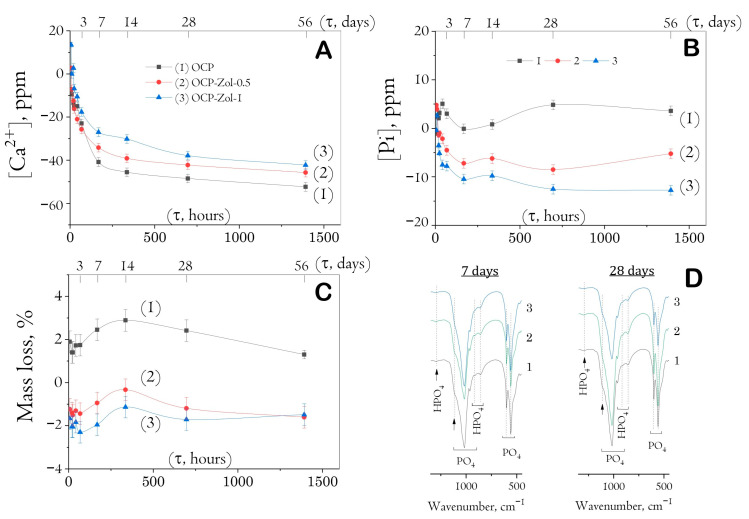
Results for study of OCP-Zol behavior in SBF. (**A**)—[Ca^2+^] concentration in SBF, (**B**)—[Pi] concentration in SBF, (**C**)—mass loss of the material during soaking to SBF, (**D**)—FTIR-spectra of the material at days 7 and 28 of exposure to SBF, 1—drug-free OCP ceramics, 2—OCP functionalized with Zol of 0.5 mg/mL, 3—OCP functionalized with Zol of 1.0 mg/mL.

**Figure 11 ijms-24-11633-f011:**
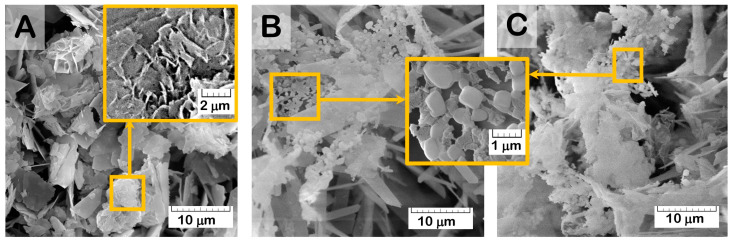
SEM images of OCP ceramics functionalized with Zol of 0 (**A**), 0.5 (**B**), and 1.0 (**C**) mg/mL, soaked to SBF for 56 days. Yellow frames show the magnified images of precipitate morphology of OCP and OCP-Zol.

**Figure 12 ijms-24-11633-f012:**
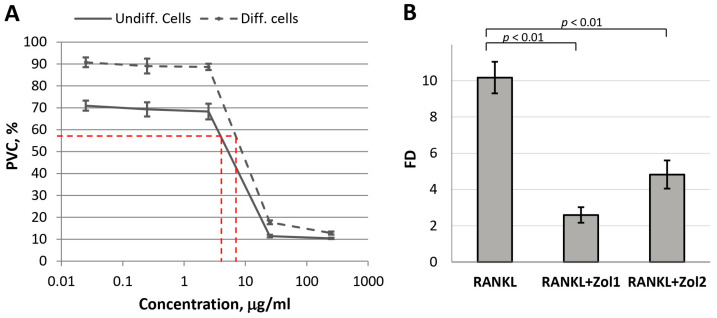
Sensitivity of RAW 264.7 cells to Zol (**A**); changes in the expression of osteoclast marker gene *TRAP* in RAW 264.7 cells upon directed differentiation with RANKL and exposure to Zol at concentrations of 0.025 µg/mL (Zol1), 0.25 µg/mL (Zol2) (**B**). Control—untreated RAW 264.7 cells, normalization gene—mouse *GAPDH*.

**Figure 13 ijms-24-11633-f013:**
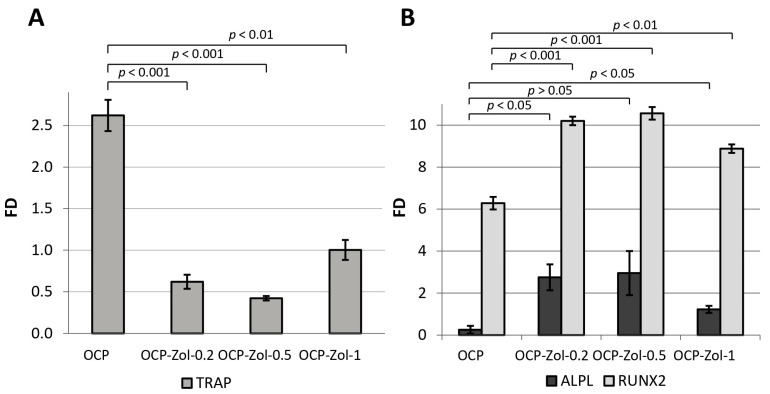
Changes in osteogenic expression in progenitor cells under the influence of OCP, depending on the content of Zol in OCP: *TRAP* expression in RAW 264.7 cells (**A**); expression of *RUNX2* and *ALPL* in BM MSCs (**B**). Control—untreated RAW 264.7 cells/BM MSCs, normalization genes—mouse and human *GAPDH*.

**Figure 14 ijms-24-11633-f014:**
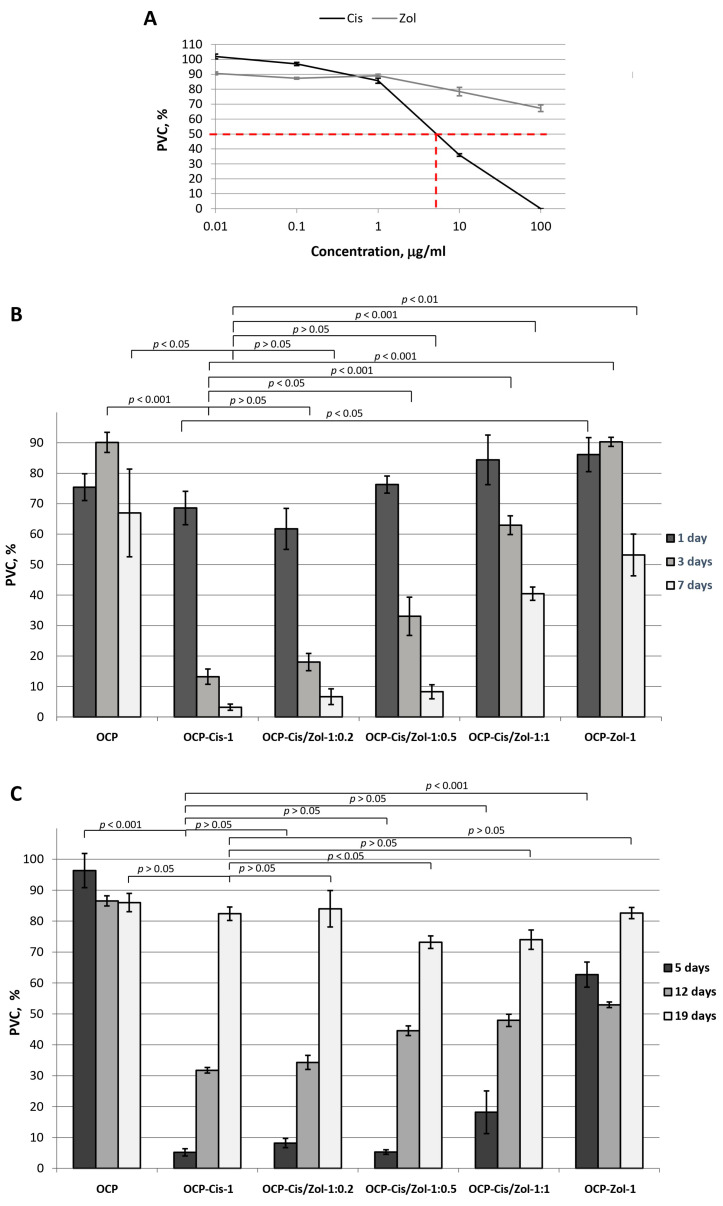
Sensitivity of MCF-7 cells to Cis and Zol (**A**); cytostatic effect of functionalized OCP ceramics on MCF-7 cells (**B**); cytostatic effect of functionalized OCP ceramics on MCF-7 cells after preliminary soaking in CGM for 5–19 days (**C**). Control—untreated MCF-7 cells.

**Figure 15 ijms-24-11633-f015:**
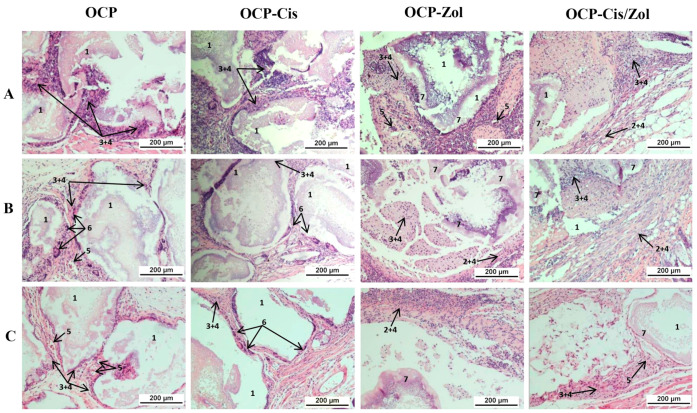
Histological analysis of subcutaneous tissue with implanted drug-free and functionalized OCP ceramics at different time points: (**A**)—4 weeks, (**B**)—8 weeks, (**C**)—12 weeks; hematoxylin-eosin staining: 1—OCP granule; 2—fibrous capsule; 3—intergranular connective tissue; 4—lymphocytic infiltration; 5—blood vessel; 6—foreign-body giant cell; 7—amorphous colloidal substance.

**Figure 16 ijms-24-11633-f016:**
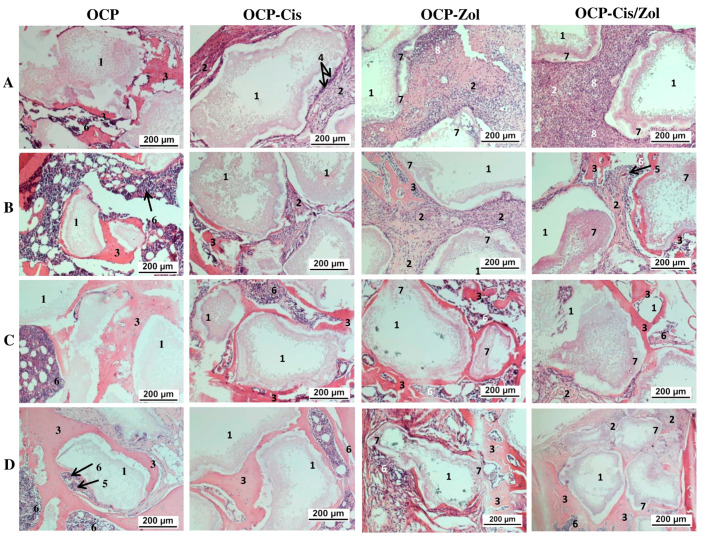
Histological analysis of rat tibia defect with implanted drug-free and functionalized OCP ceramics at different time points: (**A**)—4 weeks, (**B**)—8 weeks, (**C**)—12 weeks, (**D**)—6 months; hematoxylin-eosin staining: 1—OCP granule, 2—connective tissue, 3—bone tissue, 4—foreign-body giant cells, 5—blood vessel, 6—bone marrow hematopoiesis, 7—amorphous colloidal substance, 8—lymphocytic infiltration.

**Figure 17 ijms-24-11633-f017:**
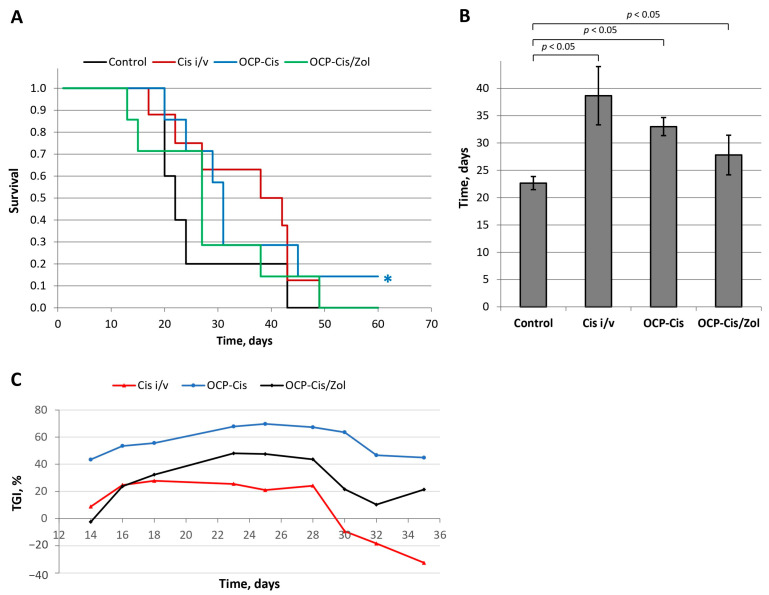
Study of the antitumor properties of functionalized OCP ceramics: (**A**)—Kaplan–Meier survival analysis, * survivor animal; (**B**)—mean life span; (**C**)—tumor growth inhibition.

**Table 1 ijms-24-11633-t001:** Description of groups of animals included in in vivo studies.

Group Number	Groups	Description	Number of Animals		
			Biocompatibility	Osteoconductive Properties	Antitumor Properties
Control	Control	Tumor tissue inoculation	–	–	5 mice
Group 1	OCP	Implantation of drug-free OCP	7 mice	9 rats	7 mice
Group 2	OCP-Cis	Implantation of OCP functionalized with Cis	7 mice	9 rats	7 mice
Group 3	OCP-Zol	Implantation of OCP functionalized with Zol	7 mice	9 rats	–
Group 4	OCP-Cis/Zol	Implantation of OCP functionalized with Cis and Zol	7 mice	9 rats	7 mice
Group 5	Cis (i/v)	Inoculation of tumor tissue simultaneously with intravenous administration of Cis	–	–	5 mice

## Data Availability

Not applicable.

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
