# Peer review of "Functionalization of Octacalcium Phosphate Bone Graft with Cisplatin and Zoledronic Acid: Physicochemical and Bioactive Properties"

_ijms, 2023, doi:10.3390/ijms241411633_

Round 1

Reviewer 1 Report

The paper ‘Functionalization of octacalcium phosphate bone graft….’ well describes one example of OCP-ceramic material functionalized with cytostatic molecules. The introduction is suitable, the study is appropriately developed, the investigation techniques are well selected and the conclusions well outlined. My overall judgment is ‘acceptable BUT with extended revision’. I suggest the Authors to take into account the following points to better present the data, because is not easy to follow the large amount of experimental issues reported.

General comments: i) the layout must be improved, place each Figure close to the text referring to it; in this journal the Authors are responsible of the pagination, so make the text easy to be read without excessive scrolling; ii) reduce considerably the number of Figures; even the journal policy don’t give any limit, now the manuscript look like a thesis or an histology atlas: select accurately the most important plots and in particular photos and provide the others as supplementary file;  iii) provide more details in the Figure captions.

Furthermore correct these points:

Is References 8 pertinent at page 2, line 55? Please verify, it seems too generic, there are more specific papers concerning ‘…escalation of therapeutic doses of drugs leads to pronounced toxic side effects’

References 9 and 21 are the same! Please correct appropriately and renumber!

Pag. 3 line 137; PVC data are non presented, ok, but please specify what was taken as control in it.

The same also for the other tests used. Since an experiment can have different significance in relation to what we are comparing, I suggest the Authors specify (may be also in the Figure Captions and/or in the exp. Section) what they used as ‘control’ for each experiment presented along the paper.

In Fig. 4B the incubation time is 2 days? Please specify it in the caption.

Page 4, line 177 and caption Fig 4C: discrepancy in the value of initial concentration in the incorporative solution (2 mg/ml in the text; 1 mg/ml in the caption).

Page 4, line 190; why the Authors do not performed XRD scans in order to detect the real nature of this new phase? In our country we ask the government stop to sell weapon, I ask the Authors not to be “neutral, but allied for peace”. Please, do the most you can for peace.

Fig. 6A and 6B: the incubation time used is 35 days for both?

Caption Figure 9A, specify release time, 5 week?

Figure 11, the yellow squares highlight the appearance of rounded particles, but where are the elongated formations as in Figure 8?? (invoked at page6, line257)

Caption Figure 12 check asterisk position:  PVC* as in axis title

Pag. 7 line 306: ….concentration range from 0.01 to 100.0 μg/ml was confirmed.  In Figure 14 it seems the 0.01 to 100 is the Log10 value of concentration……!! What is correct?

Pag 25 line 679 an inversion is present, replace with ‘…. by Forte L. et al. and Boanini E. et al. [41, 52]’

Author Response

Please see the attacment.

Reviewer 2 Report

Dear Authors,

Thank you for writing the manuscript and for your contribution in the research on the functionalization of OCP with cisplatin and zoledronic acid.

I carefully reviewed the article and the following are my suggestions for improving it.

General observations

1.     Even though there is no requirements of the journal on the maximum length of the manuscript and the number of figures I highly suggest to move some to the supplementary material. ~40 pages and especially 23 figures is quite a lot and the readers will be discouraged to use the manuscript. Please move some of the Figures to the supplementary material.

2.     Abstract is too long. Maximum of 200 words are stated and the abstract has 257. Please shorten it.

3.     I haven’t been able to see the graphical abstract, is it uploaded?

4.     Please restructure the manuscript so it is easy to follow the results. Move the figures next to the corresponding text.

5.     Please go through the manuscript and change the Latin words into italic format (i.e., in vitro and in vivo).

Materials and Methods

1.     4.2. Fabrication of OCP-ceramics – please indicate the ratio of acetate buffer and TCP granules, also indicate the time of the immersion needed to transform it fully to OCP. Which was the molarity and pH of the buffer? Were the granules mixed during the immersion or not? As it is written there is not enough details for someone to successfully repeat the experiment and get the OCP phase.

2.     4.6. Drug Release Study – please specify how much of the solution was taken out and placed back in.

3.     4.11. Evaluation of OCP-Ceramics Cytotoxicity – please indicate how were all the tested granules sterilized.

4.     4.11. Evaluation of OCP-Ceramics Cytotoxicity and 4.12. Evaluation of OCP-Ceramics Cytocompatibility – why have the authors used different concentrations and types of contact in the two assays? In the cytocompatibility the contact was direct? If it was direct the step have the granules been washed out before MTT test?

5.     4.14. Assessment of Cytostatic Properties of Functionalized OCP-Ceramics – if I understood correctly, 50 mg of granules were placed in each well that had 1 mL of media inside? Why did the authors choose such a high concentration? What is the rationale behind it?

6.     The cell density of each test has been different, all concentrations, and contact methods have been different. How do Authors explain why they chose those set ups of experiment and also this definitely impacts the results that are obtained. Can it even be compared?

7.     Please move the Table 1 to the supplementary material

Results

1.     Please move the figures next to the text describing them.

2.     1D, X axis - the unit cm-1 corresponds to the wavenumber, not wavelength.

3.     If more than one sample was measured please indicate the SD for SSA of OCP granules (The SSA of OCP-ceramics was 29.9 m2/g.)

4.     Please present the data for MTT assay in direct contact with OCP granules in % of cell viability not OD. Why were the data of indirect contact not presented? Moreover, usage of abbreviated negation terms is not supported in the official documents (aren’t should be are not).

5.     Standard deviation for 4A, B; and 6A, C and half of B is missing, please add.

6.     Considering that the SEM image 8C is showing different morphology all together, XRD is needed to show the phase present after Zol incorporation.

7.     The Authors did not include the physico-chemical observation of the OCP phase after cis and Zol absorption. I highly suggest to add XRD analysis, also after SBF immersion. FTIR only shows chemical composition, it is not enough to claim the phase change or definite effect on the OCP that is quite sensitive.

Discussion

1.     “Moreover, the higher was the content of Zol in OCP, the slower was the release of the cytostatic, the burst release phase also was less pronounced.” – how do the Authors explain this behaviour?

2.     “These data are consistent with the results of in vitro experiments, which show that upon functionalization with Zol, a new phase, different from HA, is formed on the OCP surface, slowing down the rate of ceramic dissolution.” – there is not enough data confirming this statement.

Manuscript is well written. 

Reviewer 3 Report

The manuscript entitled “Functionalization of octacalcium phosphate bone graft with cisplatin and zoledronic acid: physicochemical and bioactive properties” is very interesting and in the field of tissue engineering that is rapidly progressing. This manuscript evaluates the properties of octacalcium phosphate, calcium phosphate that is not so much investigated compared to hydroxyapatite, which is still considered a gold standard in bone tissue engineering. However, octacalcium phosphate has shown better regenerative properties than hydroxyapatite and new studies on this are highly valuable.
Below authors can find suggestions and comments that can further improve the quality of the manuscript.

The abstract is very well written.

The introduction is very well written. However, the reason why octacalcium phosphate (and not other CaP) was used should be explained in more detail. The authors could mention better regenerative properties of OCP than other calcium phosphate. This was confirmed by Suzuki et al. during their studies during the last 30 years, which were then summarized in a recent review paper entitled “The ionic substituted octacalcium phosphate for biomedical applications: A new pathway to follow?”; where also is mentioned that in our natural bone tissue, OCP is formed before HAp. The authors should improve the Introduction with more details and reasons why OCP was chosen.

The authors should better highlight the impact and novelty of this study at the end of the Introduction.

The authors did a detailed characterization of the biological properties of obtained materials.

For the reader, it will be easier to read the manuscript if the figures are placed at the proper positions in the manuscript and not at the end of the Results section.

Can histological images be lighter in color (without grey noise) so the results of tissue are more visible?

Authors should compare obtained results in this study with results already published on related topics. The authors already did this in the discussion, but it should be increased.

What are the limitations of this study?

Materials and methods are appropriately described.

Round 2

Reviewer 2 Report

The Authors have addressed the issues mentioned in the previous revision round, in a satisfactory manner. However, they have added that the sterilization of the material was done in the following way "OCP-ceramics were sterilized at 120oC for 2 hours using thermostat (Binder, Ger-1170 many).". This raises a concern regarding the OCP phase. 

Everyone working with this phase knows the sensitivity of it to the temperature - possible collapse of the water layer can happen at this high temperature and lead to the presence of a different phase in the final product. 

Due to this, please provide an XRD from the granules before and after the sterilization process. Also for the functionalized material if it has been sterilized in the same way.

Minor corrections of the newly added text are needed. This can be also fixed during the editor check if the paper will be accepted. In general, very well written. 

Round 3

Reviewer 2 Report

Thank you for your response. 

All corrections are finalized and the manuscript is ready for the acceptance.

Minor editing of English language required by the journal editor.